# The global diversity of *Haemonchus contortus* is shaped by human intervention and climate

G. Sallé [1,2]*, S.R. Doyle [1], J. Cortet[2], J. Cabaret[2], M. Berriman [1], N. Holroyd[1] & J.A. Cotton[1]

*Haemonchus contortus* is a haematophagous parasitic nematode of veterinary interest. We have performed a survey of its genome-wide diversity using single-worm whole genome sequencing of 223 individuals sampled from 19 isolates spanning five continents. We find an African origin for the species, together with evidence for parasites spreading during the transatlantic slave trade and colonisation of Australia. Strong selective sweeps surrounding the β-tubulin locus, a target of benzimidazole anthelmintic drug, are identified in independent populations. These sweeps are further supported by signals of diversifying selection enriched in genes involved in response to drugs and other anthelmintic-associated biological functions. We also identify some candidate genes that may play a role in ivermectin resistance. Finally, genetic signatures of climate-driven adaptation are described, revealing a gene acting as an epigenetic regulator and components of the *dauer* pathway. These results begin to define genetic adaptation to climate in a parasitic nematode.

[1] Wellcome Sanger Institute, Wellcome Genome Campus, Hinxton, Cambridge CB10 1SA, UK. [2] INRA - U. Tours, UMR 1282 ISP Infectiologie et Santé Publique, Centre de recherche Val de Loire, Nouzilly, France. *email: Guillaume.Salle@inra.fr

Nematodes have evolved to exploit a wide diversity of ecological niches. Although many sustain a free-living lifestyle, parasitic nematodes rely on one or more hosts to complete their life cycle. Many parasitic nematodes undergo a complex series of morphological changes, linked to migration through their hosts to establish a mature infection[1]. Their complex life cycles may involve both intermediate hosts, vectors and time spent in the environment, where they face harsh and variable conditions such as frost or drought that they must withstand between infections of their hosts[2]. Parasitic nematodes have adapted to a wide range of threats, including predation, climate and the immune responses of a great diversity of both plant and animal host species[3,4].

The evolutionary success of parasitic nematodes comes at a cost to humans, either directly as they significantly impact human health (amounting to a loss of 10 million disability-adjusted life-years)[5], or indirectly via major economic losses in plant[3] and livestock production[6], and through spending on parasite control. The control of animal parasitic nematodes relies almost exclusively on anthelmintic drugs, administered on a recurrent basis in livestock[7] and through large-scale preventative drug administration programmes in humans[8]. Although the success of such strategies was originally undeniable, the emergence of drug-resistant veterinary parasites[6], or the reported lack of efficacy in human-infective species[9], threatens ongoing control efforts for many parasitic infections. Vaccines offer an attractive alternate control strategy against these parasites: the extensive genetic diversity and immune-regulatory properties of parasites has, however, greatly hampered vaccine development[10], and although two licensed vaccines are currently available for veterinary purposes[11], transcriptomic plasticity of the parasite following vaccine challenge may contribute to circumvent the vaccine response of their host[11]. It is therefore clear that novel, sustainable control strategies are required. The potential of helminths to adapt to and thus escape control measures lies in their underlying genetic diversity. A greater understanding of the extent of this diversity and the processes that shape it throughout their range should provide insight into the mechanisms by which they adapt, and may identify new targets, which may be exploited for control.

The trichostrongylid *Haemonchus contortus* is a gastrointestinal parasite of ruminants in tropical and temperate regions throughout the world, and causes significant economic and animal health burden particularly on sheep husbandry. It is also emerging as a model parasitic nematode system for functional and comparative genomics, largely owing to its rapid ability to acquire drug resistance, the relative tractability of its life cycle under laboratory conditions[12], the development of extensive genomic resources[13,14], and its relatively close relationship with other clade V parasitic nematodes of both veterinary and medical importance (i.e, other gastrointestinal nematodes of livestock and human hookworms)[15]. We use whole genome sequencing of 223 individual *H. contortus* sampled from 19 isolates spanning five continents to characterise genome- and population-wide genetic diversity throughout its range. This survey of genome-wide diversity reveals old and new genetic connectivity influenced by human history, and signatures of selection in response to anthelmintic exposure and local climatic variation.

## Results

### *H. contortus* isolates are genetically diverse.
Whole-genome sequencing of 223 individuals from 19 isolates (Fig. 1; Supplementary Data 1) revealed 23,868,644 SNPs with a genome-wide distribution of 1 SNP per 9.94 bp on average.

Only a proportion of the filtered SNPs were called in more than half the individuals ($n = 3,338,155$ SNPs), with only 411,574 SNPs segregating with a minor allele frequency (MAF) > 5% in those individuals. Estimates of nucleotide diversity ($\pi$) in isolates with at least five individuals ranged from 0.44% (STA.2) to 1.3% (NAM; Supplementary Table 1). Variance in $\pi$ across-autosomes among isolates was partly explained by isolate mean coverage ($F_{(1,84)} = 9.49$, $P = 0.003$; Supplementary Note 1). To account for this bias, $\pi$ was estimated for three subsets of isolates with the highest coverage ($> 8\times$ on average) from France (FRA.1 and FRA.2), Guadeloupe (GUA), and Namibia (NAM) that yielded slightly higher values that ranged between 0.65% and 1.14%. Overall, these data show equivalent diversity levels to *Drosophila melanogaster* (ranging between 0.53% and 1.71%)[16] but represented ~ 1.6- to 45-fold greater diversity than two filarial nematode species for which similar statistics are available (0.02% for *Wuchereria bancrofti* larvae;[17] 0.1% and 0.4% for $\pi_S$ and $\pi_N$ in *Onchocerca volvulus*[18] Supplementary Fig. 1).

To begin to explore the global diversity of *H. contortus*, we performed a principal component analysis (PCA) of genetic variation, which revealed three broad genetic clusters of isolates (Fig. 2a) that largely coincided with the geographic region from which they were sampled, including: (i) Subtropical African isolates (NAM, STA, and ZAI), (ii) Atlantic isolates including Morocco (MOR), São Tomé (STO), Benin (BEN), Brazil (BRA) and Guadeloupe (GUA), and (iii) the remainder from the Mediterranean area (FRA, ACO) and Oceania (AUS.1, AUS.2 and IND). The greatest diversity among samples was identified in the African and South American isolates, with East African samples spread along PC1 and West African and South American samples along PC2.

Using estimates of nucleotide diversity, together with the *C. elegans*[19] mutation rate and assuming a balanced sex ratio[20], we inferred the current effective population size ($N_e$) of *H. contortus* to be between 0.60 and 1.05 million. MSMC analysis, which models past recombination events based on heterozygosity patterns along the genome, revealed the historical $N_e$ has remained within a slightly lower range of values for most of the sampled time interval, i.e., from 2.5 to 500 thousand years ago (kya), with extreme estimates falling between $1.5 \times 10^5$ and $6.1 \times 10^5$ individuals for GUA (633 years ago) and NAM (415 years ago) isolates, respectively (Fig. 2b). $Ne$ estimates remained relatively constant in most populations until 2.5 Kya; since this time, GUA and FRA.1 isolates suffered a more drastic reduction in $N_e$, which ranged for each isolate between $0.859 \times 10^5$ and $1.2 \times 10^5$ individuals ~ 633 and 452 years ago, respectively (Fig. 2b).

**Population connectivity of *H. contortus*.** To explore the global connectivity between isolates, we used a number of complementary approaches. Phylogenetic relationships determined by nuclear (Supplementary Fig. 2) and mitochondrial (Fig. 3a; PCA of mtDNA genetic diversity is presented in Supplementary Fig. 3) diversity broadly supported the initial PCA analysis (Fig. 2a), each revealing three main groups of samples partitioned by broad geographic regions, i.e., Africa, Oceania and Mediterranean groups. However, the mitochondrial data revealed further subdivision of the Oceanian and Mediterranean clades not present in nuclear data alone.

The presence of close genetic relationships between geographically distant isolates, resulting in a weak phylogeographic signal (Mantel's test $r = 0.10$, $P = 0.001$), was inconsistent with a simple isolation-by-distance scenario. This observation was supported by, for example, little genetic differentiation (measured by $F_{ST}$) between geographically distant French (FRA) and Oceanian (AUS.1, AUS.2, IND) isolates (Fig. 3b).

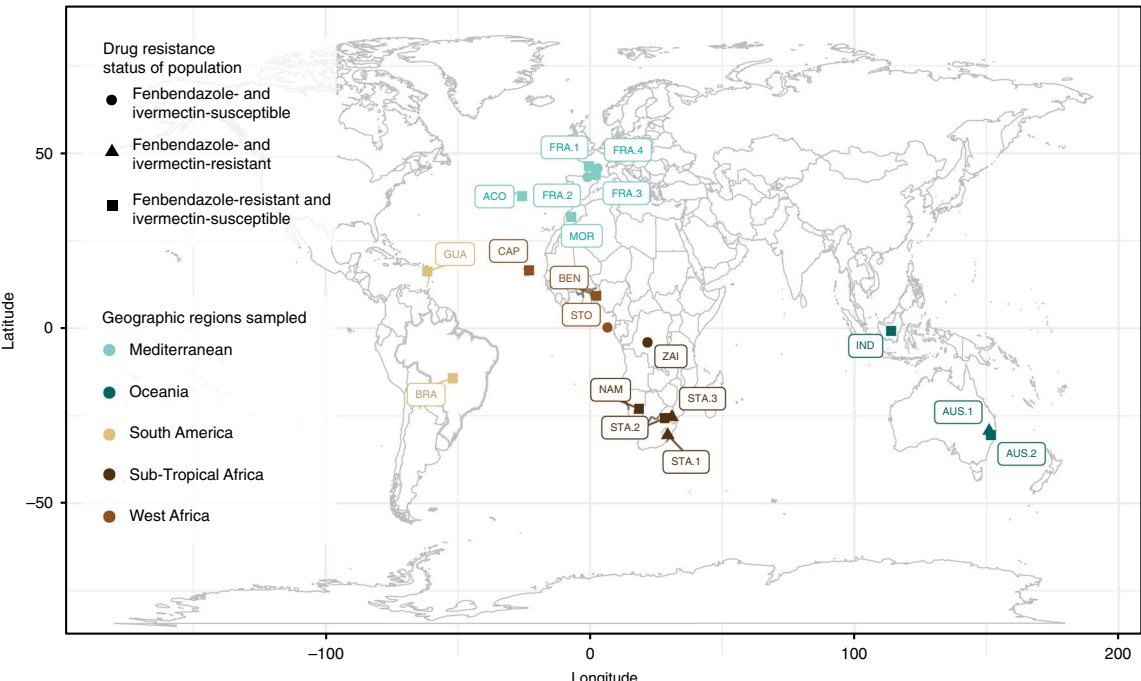

**Fig. 1** Global distribution of *Haemonchus contortus* isolates. Isolates sampled are coloured by geographical region (sand: South-America, brown: Western-Africa, light green: Mediterranean area, dark brown: Subtropical Africa, dark green: Australia). Shape indicates the anthelmintic resistance status of each isolate (susceptible = circles; resistant to fenbendazole = squares; resistant to both ivermectin and fenbendazole = triangles)

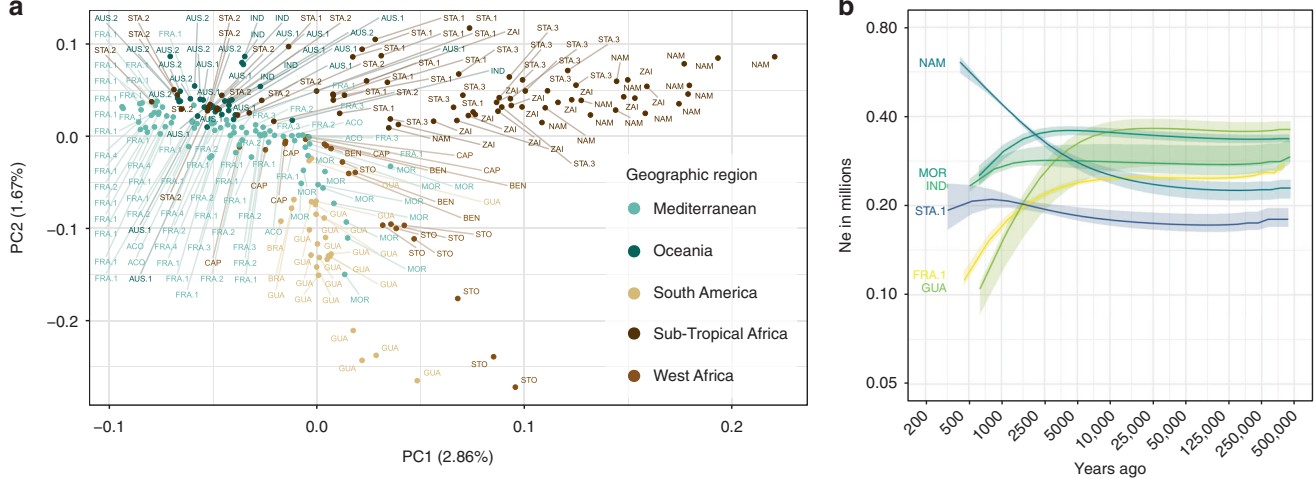

**Fig. 2** Global diversity of *Haemonchus contortus* isolates. **a** Principal component analysis of genomic diversity. It is based on genotype likelihood inferred from whole-genome sequences of 223 individual males (243,012 variants considered). Samples are coloured by geographic region described in Fig. 1. **b** MSMC effective population size across time for six isolates. Coloured shaded area represents range of values estimated from a cross-validation procedure with five replicates, computed by omitting one chromosome at a time

The greatest genetic dissimilarity was found between sub-tropical African isolates and the French and Australian isolates, with mean $F_{ST}$ across comparisons of 0.21 ($F_{ST}$ range according to isolate pairs: 0.06–0.42) and 0.24 ($F_{ST}$ range: 0.17–0.33), respectively (Fig. 3b, Supplementary Fig. 4). The divergence of African isolates was likely owing to the higher within-isolate diversity (+19% higher mitochondrial nucleotide diversity) relative to others ($t$ value = 3.761, d.f. = 1227, $n$ = 1232, $P$ = 0.002): windowed mitochondrial nucleotide diversity estimates were 0.63% ± 0.37% (standard deviation), 0.58% ± 0.31%, 0.66 ± 0.32%, 0.6% ± 0.35%, for Mediterranean ($n$ = 360), Oceanian ($n$ = 225), American ($n$ = 92), and south-African ($n$ = 305) isolates, respectively.

The higher genetic differentiation (8.9% difference, $F_{(1,134)}$ = 11.36, $P$ = 0.0009) between STO and other isolates relative to other pairwise comparisons, almost certainly reflects the isolation of an island population relative to continental populations as a result of higher exposure to drift or smaller founding populations (Fig. 3b). STO samples also displayed higher genetic dissimilarity (5.7% difference, $F_{(1,7567)}$ = 581, $P < 10^{-4}$) to other samples. Inference of the joint history of STO and other African populations supported this view. The best-fitting model (Supplementary Data 2) was consistent with either ancient symmetrical gene flow followed by isolation or an early split followed by secondary contact in late 1800's before isolation with MOR population (Supplementary Data 2). This latter demography

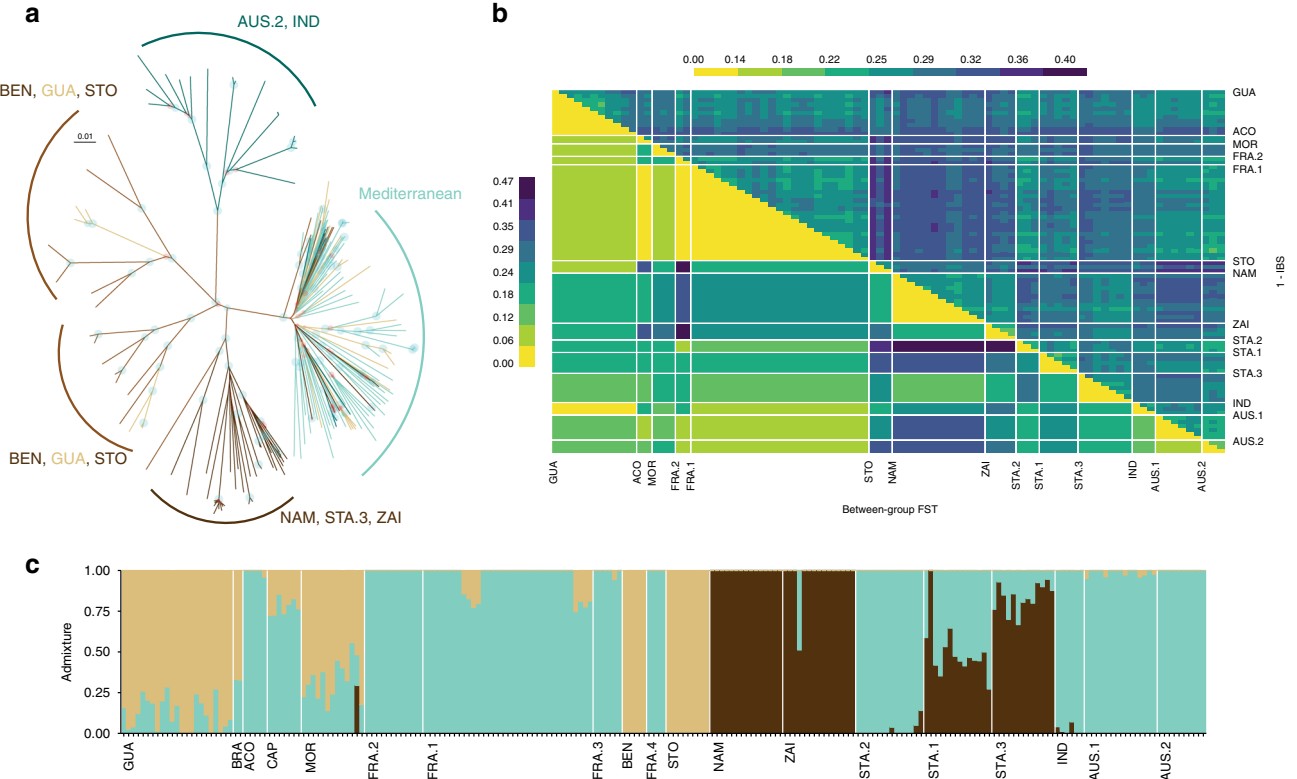

**Fig. 3** Global connectivity of *Haemonchus contortus* isolates. **a** Unrooted maximum likelihood phylogenetic tree of 223 mitogenomes. The tree was constructed using 3052 SNPs from 223 individual mitochondrial genomes. Circles indicate bootstrap support for each node, blue if support was higher than 70%, red elsewhere. Branches leading to a sample identifier are coloured by geographic region described in Fig. 1. Mitogroups are annotated with constitutive sample populations. **b** Genetic dissimilarity and divergence among isolates. The matrix shows pairwise dissimilarity between individuals (upper right) and pairwise $F_{ST}$ between isolates (lower left). **c** Admixture analysis of 223 individuals. A cluster size of $K = 3$ is presented, determined from sites with minor allele frequency above 5% and call rate higher than 50%. Admixture pattern for other values of K is provided in Supplementary Figs. 5. Isolates are presented sorted along their longitudinal range, and samples sorted by assignment to the three coloured clusters

would underpin the pattern of admixture detected between STO and MOR (Fig. 3c).

A closer inspection of GUA samples revealed a mixed ancestry, likely derived from West African and Mediterranean heritage. A subset of GUA samples showed limited genetic differentiation to FRA isolates ($F_{ST}$ range = 0.07–0.10; Fig. 3b), whereas the remaining GUA samples were genetically similar to isolates from the West African coast, i.e., ACO, BEN, MOR, STO (Fig. 3b, c), as indicated by lower $F_{ST}$ estimates with these isolates ($F_{ST}$ range = 0.07–0.13; Wilcoxon's test, $P = 0.07$; Fig. 3b) and evidence of shared ancestry from the admixture analysis (Fig. 3c, Supplementary Figs. 5 and 6). A particularly close relationship was identified between STO and GUA ($F_{ST} = 0.10$; Fig. 3b). This conflicting genetic origin among GUA sub-isolates is responsible for the higher nucleotide diversity observed in GUA as a whole (Supplementary Table 1).

The patterns of genetic connectivity support at least three distinct migration events in time and space that we investigated using forward genetic simulations (Supplementary Data 2). First, we detect an out-of-Africa scenario, whereby the greatest diversity was sampled within Africa, and that isolates outside of Africa represent a subset of this diversity. Consistent with this hypothesis, nuclear genome variation of non-African isolates experienced a genetic bottleneck that occurred between 2.5 and 10 kya that is not present in the African isolates sampled (Fig. 2b). Bayesian coalescent estimate of this initial divergence from mitochondrial genome data yielded an overlapping time range of between 3.6 and 4.1 kya (Supplementary Fig. 7). Genetic simulations of the joint demography between FRA.1 and African

populations also favoured complex scenarios involving early split (maximum likelihood estimates ranging between 13 and 25 Kya) followed by ongoing gene flow with STA populations or more recent isolation with Namibia (Supplementary Data 2). Second, the genetic connectivity of West African and American isolates is consistent with parasites spreading during the trans-Atlantic slave trade movement. The scenario linking GUA and STO was compatible with initial population division occurring around 1640 Common Era (CE) ± 167 years ($n = 100$; Supplementary Data 2), consistent with migration associated with colonisation and slave trade that occurred in the West Indies under French influence during that time[21]. The third pattern of connectivity likely reflects British colonisation of Australia in late 1700's: the interweaving of Australian and South-African worms into the Mediterranean phylogenetic haplogroup (Fig. 3a, b) may mirror the foundation of Australian Merino sheep, which were first introduced into Australia from South-Africa, before additional contributions from Europe and America were made[22]. The admixture pattern observed for worms from the two countries matched their shared ancestry (Fig. 3c), as well as a genetic connection between America and Australia (seen for the AUS.1 isolate; Fig. 3c). Although maximum likelihood estimates supported these scenarios with the isolation of European and South-African isolates occurring between 1794 and 1871 CE (Supplementary Data 2), wide confidence intervals for these estimates limited our ability to specifically define the timings of these events (Supplementary Data 2). However, the split between Australian isolates occurred in 1895 CE ± 132 years ($n = 100$), consistent with the initial foundation of the sheep industry in this

country. These complex patterns reiterate that human movement has played an important role in shaping the diversity of this livestock parasite throughout the world.

**Drug resistance shapes diversity in the genome.** The extensive use of anthelmintics has been and remains the primary means of control of *H. contortus* and other gastrointestinal nematodes worldwide. This strategy has resulted in the independent emergence of drug-resistant isolates throughout the world, which now limits farming in some areas. Strong selection should impact the distribution of genetic variation within isolates; signatures of selection may reveal genes associated with drug resistance, knowledge of which may contribute to monitoring the emergence and spread of drug-resistant isolates, and the design of new control strategies.

The genetic determinants of benzimidazole resistance is perhaps the best characterised of all anthelmintics, with any one of three amino-acid residue changes (F167Y, E198A and F200Y) in the beta tubulin isotype 1 (*Hco-tbb-iso-1*) protein capable of mediating phenotypic resistance. We identified indications of a selective sweep in the region surrounding the *Hco-tbb-iso-1* locus in the resistant isolates analysed. Focusing on three isolates with the highest coverage, we found an average 2.31-fold reduction of Tajima's *D* coefficient within 1 Mbp (Fig. 4a, Supplementary Fig. 8) and an average 33% reduction in nucleotide diversity (Supplementary Fig. 9) within this region relative to the rest of chromosome I.

This signature was most evident in the GUA and NAM isolates, but was weaker in the French isolate (FRA.1) owing to both phenotypically susceptible and resistant individuals being present (Supplementary Table 2). A phylogenetic network based on genotype information over the whole *Hco-tbb-iso-1* locus (Supplementary Fig. 10) revealed that resistant populations were polyphyletic, suggesting resistant mutations have generally evolved independently in different populations, e.g., Namibia (NAM) and South-Africa (STA.1, STA.3) and to a lesser extent in Australia. However French and Guadeloupian individuals had little divergence in their haplotypes, suggesting that gene flow of resistant haplotypes between mainland France and the West Indies may have occurred (Supplementary Fig. 10). A topology analysis of a 100 Kbp region spanning the *Hco-tbb-iso-1* locus supported the shared phylogenetic origin between FRA.1 and GUA (topology 3), whereas the surrounding region was in favour of topologies congruent with overall population structure (Fig. 4b). This finding is consistent when either Moroccan (Fig. 4b) or São Tomé populations are used as a population with close ancestry with GUA (Supplementary Fig. 11).

We analysed the frequency of the three well-characterised resistance-associated mutations affecting codon positions 167 (T to A), 198 (A to T) and 200 (T to A) (Supplementary Table 2; Supplementary Table 3; Supplementary Figs. 12 and 13).

The F200Y homozygous genotype was the most common and widespread resistant genotype ($n = 39$), and accounted for all samples from the Guadeloupe ($n = 14$) and the White-River South-African (STA.3; $n = 6$) isolates. Variants at codons 167 and 198 were much less common, i.e., 13 individuals carried mutant alleles at position 198 and only one F167Y mutant was identified in a French isolate. No double homozygous mutants were found, but three individuals from France, Australia and South-Africa were heterozygous at both positions 198 and 200. In each case, inspection of the sequencing reads revealed that the two mutations never appeared together in the same sequencing read, suggesting they cannot co-occur in *cis* (on the same chromosome copy). Genotype frequencies were consistent with isolate-level benzimidazole efficacies as measured by the percentage of egg

excretion after treatment (Supplementary Table 2). Samples analysed from suspected susceptible isolates always presented with the susceptible genotype for each of the three positions considered ($n = 20$).

The genetic determinants of ivermectin resistance are largely unknown, but many genes have been proposed to be associated with resistance; although one interpretation of this observation is that ivermectin resistance is a multigenic trait, a major locus associated with resistance in Australian and South-African *H. contortus* has recently been mapped to a region ~37–40 Mbp along chromosome V[13]. To examine for the presence of ivermectin-mediated selection in our data, a pairwise differentiation scan was performed between known ivermectin-resistant isolates and other isolates sharing same genetic ancestry, i.e., STA.1 and STA.3 vs. NAM and ZAI in Africa, AUS.1 against AUS.2 in Australia (Fig. 5a).

The previously described region on chromosome V[13] appeared differentiated, particularly between pairwise comparisons involving the South-African resistant isolate (Fig. 5a). A second major differentiation hotspot spanned a 3 Mbp region of chromosome I and encompassed the *Hco-tbb-iso-1* locus (Fig. 5a). Although non-synonymous mutations at codon positions 167, 198 and 200 of the β-tubulin isotype 1 are associated with resistance to benzimidazoles, it has been proposed that there may also be an association between this gene and ivermectin resistance[23,24]. Although our data seem to support this association, an attempt to narrow-down the differentiation signal in this region found significant differentiation in only two 10-kbp windows in two comparisons (NAM vs STA.1 and NAM vs. STA.3, and these two windows did not overlap the *Hco-tbb-iso-1* locus. Furthermore, the fact that all of our ivermectin-resistant isolates were also benzimidazole-resistant suggests that this signal is confounded; pairwise $F_{ST}$ analyses between ivermectin-resistant and -susceptible isolates from the field will always be biased toward strong differentiation around the *Hco-tbb-iso-1* locus owing to loss of diversity in benzimidazole-resistant isolates. As such, it was not possible to confirm the putative association between the *Hco-tbb-iso-1 locus* and ivermectin resistance. We note that the lack of QTL evidence in this region from controlled genetic crosses using ivermectin selection[13] supports the conclusion that standing genetic variation at the *Hco-tbb-iso-1* locus is unlikely to be directly influenced by ivermectin.

To further investigate more subtle signatures of selection, we computed the XP-CLR coefficient, which simultaneously exploits within-isolate departure from neutrality and between-isolate allele frequency differences[25]. This analysis relies on called genotypes rather than genotype likelihoods (GLs), but is robust to SNP uncertainty[25]. Within continent pairwise comparisons of African (NAM, MOR, STA.1, STA.3, ZAI) and Australian isolates (AUS.1, AUS.2) yielded 1740 hotspots of diversification, 48% of which were contained within a gene locus (Supplementary Fig. 14, Supplementary Data 3). Among these hotspots, two known candidate genes associated with ivermectin resistance were identified, namely an ivermectin sensitive glutamate-gated chloride channel[26] (*HCOI00617300*, *glc-4* ortholog) on chromosome I, and a P-glycoprotein coding gene already involved in ivermectin susceptibility in the equine ascarid *Parascaris sp*[27], (*HCOI00233200*, *pgp-11* ortholog) on chromosome V. Although the former showed indication of reduced genetic diversity in its vicinity in resistant isolates relative to others ($0.48\% \pm 0.09\%$ difference in nucleotide diversity between the two groups, $n = 60$, $t$ value $= 5.25$, d.f. $= 56$, $P < 10^{-4}$; Fig. 5b), the genetic diversity pattern in the 100 Kbp window surrounding the latter was similar across isolates ($t$ value $= 0.004$, d.f. $= 56$, $P = 0.99$, $n = 60$; Fig. 5b), suggesting its role may not be specifically related to ivermectin resistance.

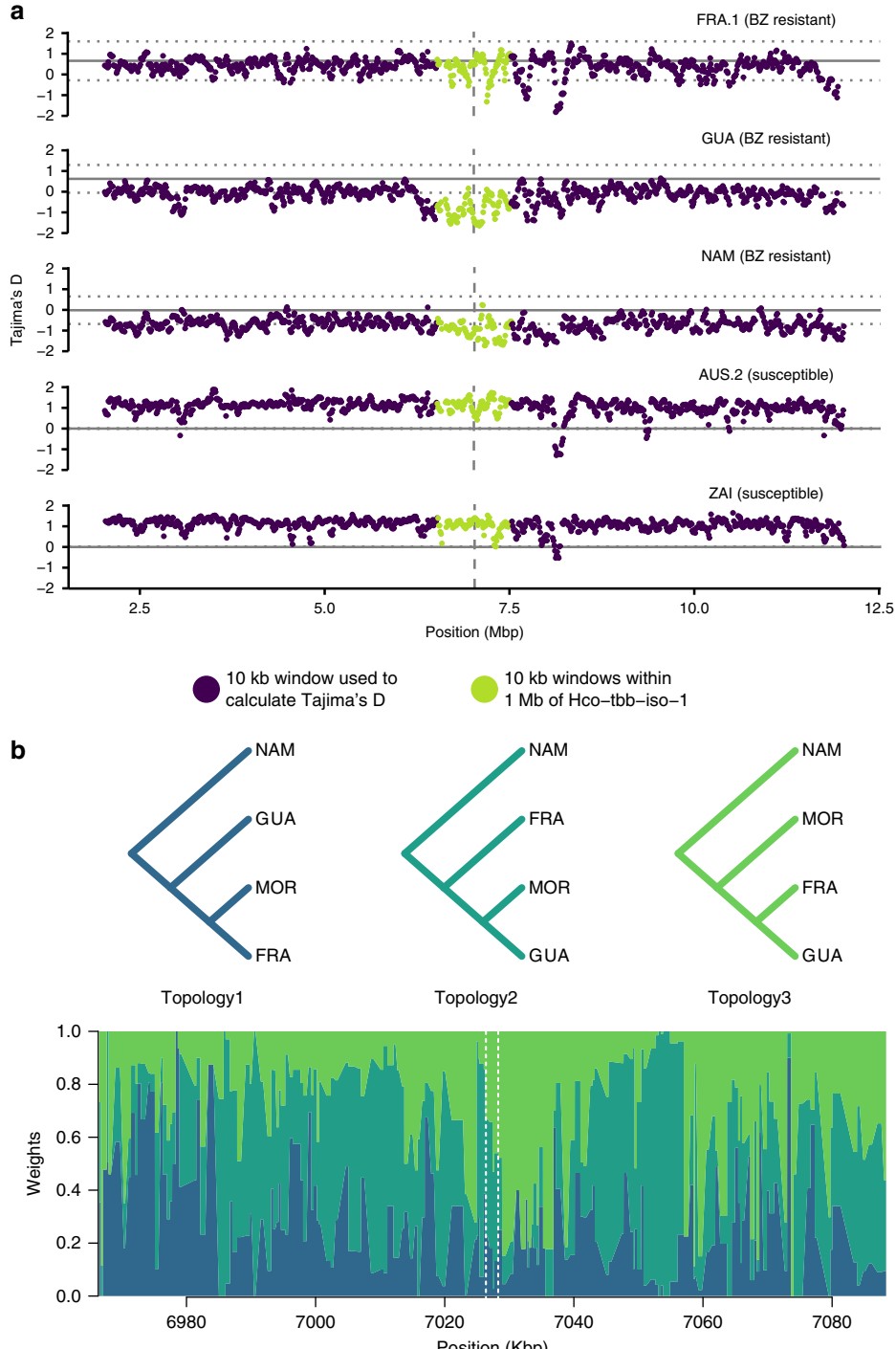

**Fig. 4** Exposure to benzimidazoles leaves a major genomic footprint. **a** Tajima's *D* surrounding the *Hco-tbb-iso-1* locus on chromosome I. A total of 10-Mbp surrounding was considered (purple), of which 1 Mbp nearest to *Hco-tbb-iso-1* is highlighted (green). Isolates compared included benzimidazole-resistant French (FRA.1), Guadeloupian (GUA), Namibian (NAM) or benzimidazole susceptible Australian (AUS.2), and Zaire (ZAI) isolates. Mean expected Tajima's *D* (solid grey line) and 99% confidence interval (dotted line) were estimated from a 1000 simulated 10-Kbp wide sequences following MSMC-inferred demography. **b** Topology weighing analysis of a 100-Kbp window centred on *Hco-tbb-iso-1*. At each position, the weight of each of the three possible topologies inferred from 50 Kbp windows is overlaid. Topology 2 (dark green) corresponds to an isolation-by-distance history, whereas topology 3 (light green) would agree with shared genetic material between worm isolates from French mainland into Guadeloupe. The *Hco-tbb-iso-1* locus is indicated by vertical dashed lines

GO term enrichment analysis of all XP-CLR significant regions identified 26 significant terms (Supplementary Table 4). The top 10 most-significant GO terms encompassed enzymatic-related activity, neurotransmitter transporter activity (GO:0005326, Kolmogorov–Smirnov test, $P = 5.8 \times 10^{-4}$) and

response to drug (GO:0042493, Kolmogorov–Smirnov test, $P = 4.3 \times 10^{-3}$). Additional significant biological process associated terms were related to phenotypes tightly linked to anthelmintic effects. For example, pharyngeal pumping[28] (GO:0043051, Kolmogorov–Smirnov test, $P = 5.8 \times 10^{-3}$) and oviposition

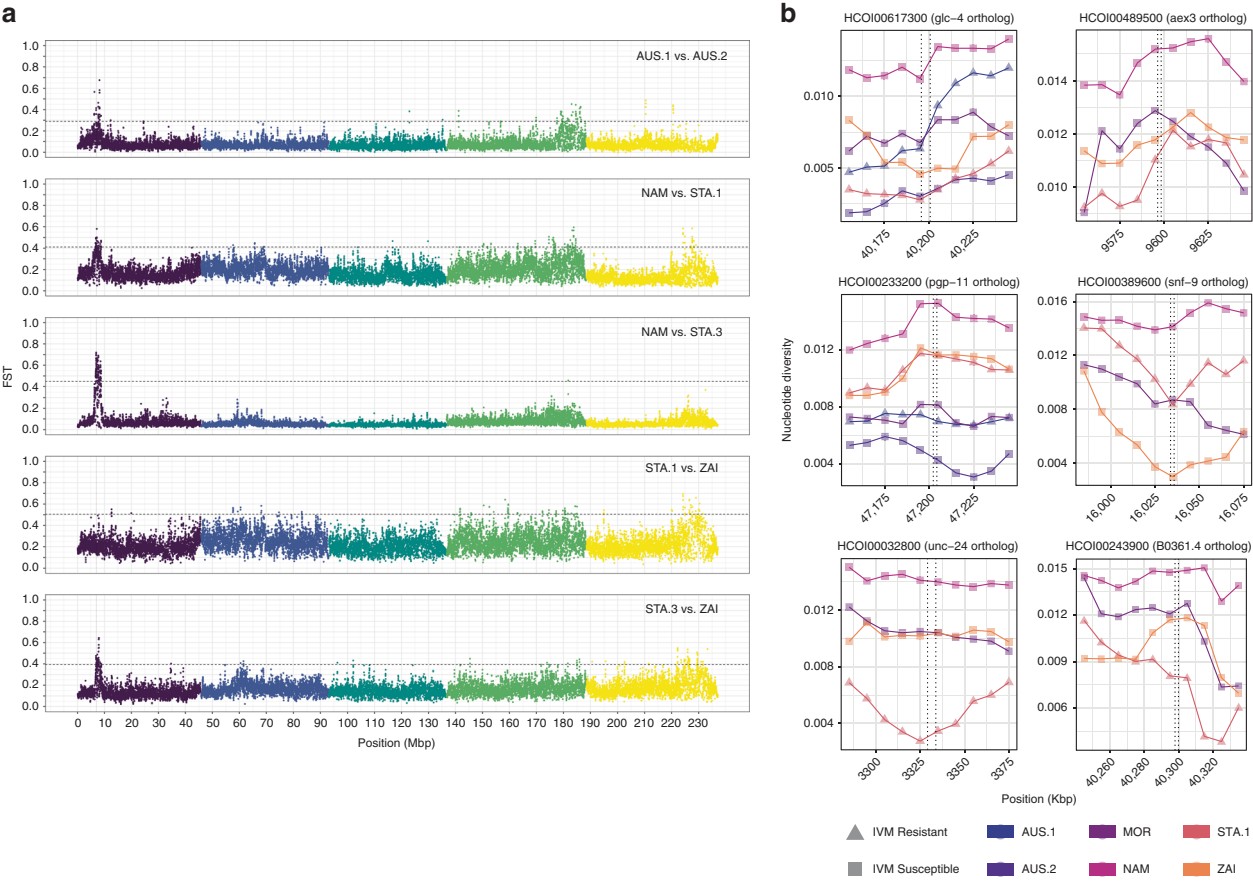

**Fig. 5** Looking for ivermectin-associated genes. **a** Genome-wide differentiation scan for ivermectin resistance. $F_{ST}$ estimates between pairs of ivermectin-resistant (AUS.1, STA.1, STA.3) and isolates with no evidence for ivermectin resistance is plotted along genomic position. Chromosomes are coloured and ordered by name, i.e., from I (purple) to V (yellow). Horizontal line represents the 0.5% $F_{ST}$ quantile cutoff. Vertical line on chromosome I points at *Hco-tbb-iso-1* locus. **b** Nucleotide diversity over candidate genes for ivermectin resistance. A 100-Kbp window surrounding the candidate gene position (highlighted by vertical dashed lines) is shown. Dots represent the mean $F_{ST}$ over a 10-Kbp window. Ivermectin-resistant isolates (IVM-R) appear as triangles and squares indicate susceptible isolates (IVM-S)

(GO:0046662, Kolmogorov–Smirnov test, $P = 8.6 \times 10^{-3}$) genes were enriched, both of which are phenotypes linked to ivermectin and its effect on parasite fecundity[29]. Neurotransmission is the primary target of anthelmintics such as macrocyclic lactones[30] and levamisole[31]; genes with GO terms related to neurotransmission (GO:0001505, Kolmogorov–Smirnov test, $P = 9.1 \times 10^{-3}$) were significantly enriched across comparisons. Anthelmintic-associated GO terms including "response to drug", "regulation of neurotransmitter" and "neurotransmitter transporter activity" were significantly enriched in every comparison involving a South-African multi-resistant field isolates (STA.1, Supplementary Data 4). Four candidate genes (HCOI00389600, HCOI00032800, HCOI00243900, HCOI00489500) were found overlapping XP-CLR signals of selection and thus seem candidates for drug resistance loci in *H. contortus*, as supported by the functions of their *C. elegans* orthologs (*snf-9*, *unc-24*, *B0361.4*, *aex-3*, respectively). The *snf-9* gene encodes a neurotransmitter:sodium symporter belonging to a family of proteins involved in neurotransmitter reuptake, and the latter three genes are expressed in neurons. *Unc-24* and *B0361.4* are involved in response to lipophilic compounds and *aex-3* is critical in synaptic vesicle release[32]. A reduction in nucleotide diversity surrounding *unc-24*, *B0361.4*, *aex-3* in the STA.1 isolate in comparison with others (Fig. 5b) may reflect evidence of drug selection.

**Climate adaptation shapes genomic variation between isolates.** In addition to putative anthelmintic-related GO terms, response to stress (GO:0006950, Kolmogorov–Smirnov test, $P = 5.9 \times 10^{-3}$) was among the top 10 significant biological process ontologies associated with genes under diversifying selection. Although anthelmintic exposure would be associated with significant stress on susceptible (and perhaps resistant or tolerant) parasites, all free-living stages of *H. contortus* will be exposed to and must tolerate abiotic factors such as temperature or humidity prior to infection of a new host. To evaluate the impact of such climatic stressors, a genome scan for genetic differentiation between isolates categorised by the climatic conditions prevailing at their sampling locations, i.e., arid (Namibia), temperate (France mainland) and tropical (Guadeloupe), was performed (Fig. 6a).

A major signal of differentiation formed of two windows (6.925–6.995 Mbp and 7.165–7.205 Mbp on chromosome I) was shared by all pairwise comparisons. Additional comparisons between populations with different histories, i.e., Australia, Indonesia, São Tomé, and Zaire also supported this signal (Supplementary Figure 15), suggesting it was independent of the precise choice of population included. Six genes were found within these two regions (Supplementary Data 5), among which orthologs of *C. elegans* genes *hprt-1*, *cpb-1*, *B0205.4* were identified. A highly differentiated 260 Kbp region of chromosome III also repeatedly occurred across comparisons. A window

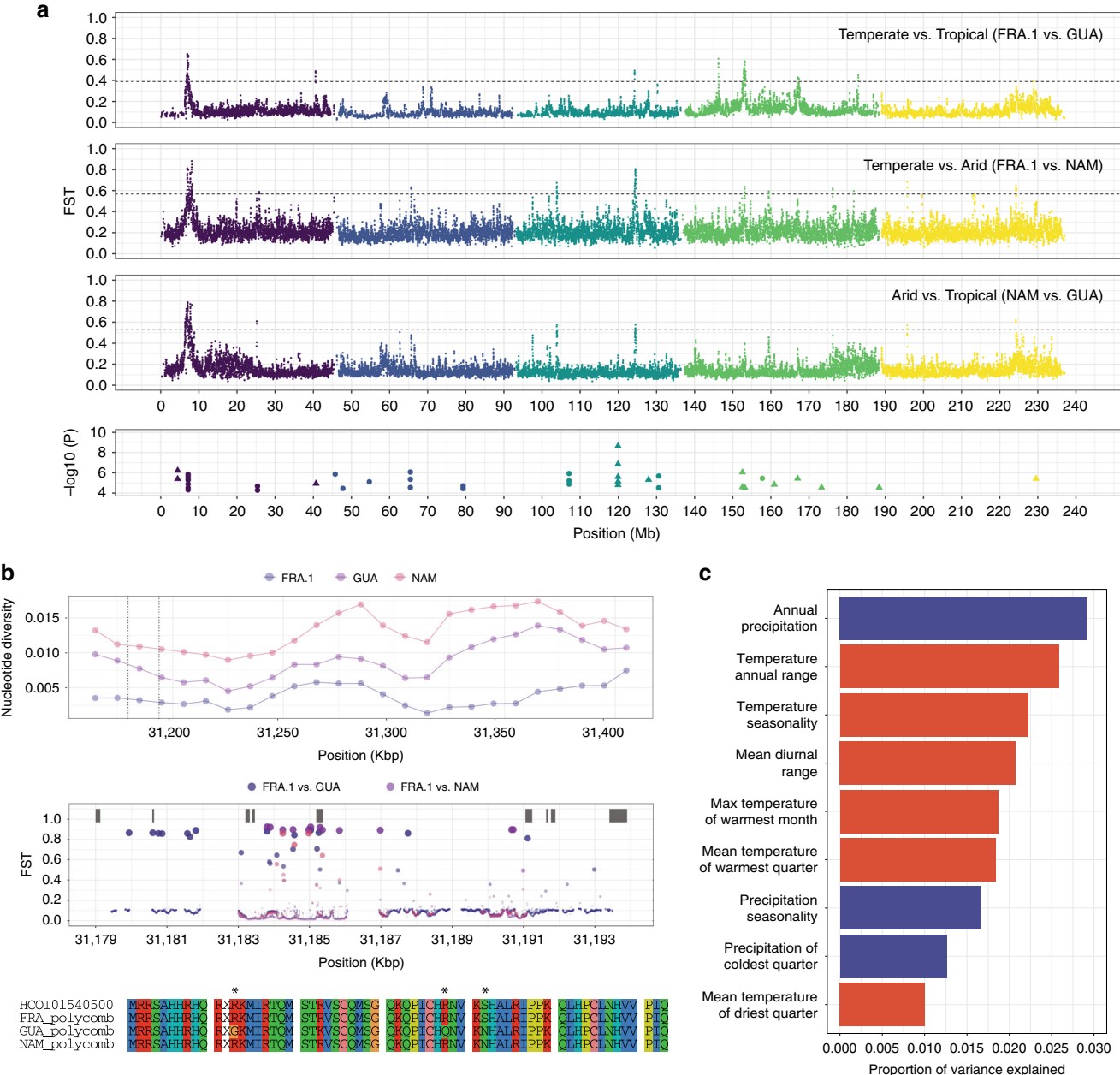

**Fig. 6** Genomic signal of climate adaptation. **a** Top three panels represent $F_{ST}$ values plotted against genomic position (Mbp), coloured and ordered by chromosome name (from I in purple to V in yellow). Horizontal dashed line represents the 0.5% quantile. Bottom panel shows genomic positions of significant associations between annual precipitation (circle) and temperature annual range (triangle). In that case, $-\log10(P)$ refers to log-transformed $P$ values estimated by the Latent Factor Mixed Model analysis. **b** A Polycomb group protein coding gene (*Pc*) underpins major differentiation signal across climatic conditions. Top panel shows nucleotide diversity estimates for 10-Kbp genomic windows spanning windows of chromosome III with high genetic differentiation between populations from arid, temperate and tropical areas. Dashed lines indicate the *Pc* ortholog boundaries. Within these boundaries, $F_{ST}$ estimates along every base-pair of the gene sequence are represented below (circle size proportional to $F_{ST}$ coefficient), with predicted exon model shown as grey rectangles. The bottom panel shows translated consensus sequences of exon 5 for populations of interest with asterisks marking mutations. **c** Proportion of genetic variance explained by climatic variables. Variants from eight isolates with no record of ivermectin resistance (AUS.2, FRA.1, GUA, IND, MOR, NAM, STO, ZAI) were analysed against a set of nine variables (from a total of 19, either precipitation- or temperature-associated coloured in blue and red respectively), selected to minimise correlation between variables. Annual precipitation (BIO12) and temperature annual range (BIO7) are main contributors of genetic variance

between 31.155 Mbp and 31.185 Mbp was common to comparisons involving isolates from arid areas (NAM vs. FRA.1 and NAM vs. GUA), and a second window (31.345–31.415 Mbp) was common to comparisons between tropical isolates and others (GUA *vs.* FRA.1 and GUA *vs.* NAM). Although the

annotated genes in the latter region were not associated with any biological description, the first window overlapped a chromo-domain containing protein (*HCOI01540500*; Supplementary Data 5). This gene is an ortholog of *Pc* (*FBgn0003042*) in *D. melanogaster*, a chromo-domain subunit of the Polycomb

PRC1-complex that specifically recognises trimethylated lysine 27 of histone 3 (H3K27me3)[33]. Nucleotide diversity over these genomic windows revealed reduced genetic diversity in FRA.1 relative to isolates from hotter climates (Fig. 6b). The nucleotide diversity pattern downstream from HCOI01540500 (31.225–31.350 Kbp, Fig. 6b) found between GUA and NAM certainly contributed to higher $F_{ST}$ in these windows. Additional differentiation analyses performed at the gene level with base-pair resolution highlighted a few discrete locations of elevated $F_{ST}$ common to every comparison and overlapping intron 4 and exon 5 of this gene (Fig. 6b). Translated consensus exon 5 sequences revealed the highest divergence in the GUA isolate (92.1% identity with reference sequence; Fig. 6b), characterised by multiple amino-acid changes whose putative functional consequences remain unknown.

To further explore the impact of climatic conditions on genetic diversity, we used a random forest based statistical approach to quantify the relationship between genetic variation and specific bioclimatic variables, describing the environment each isolate was obtained from. Bioclimatic variables were derived from monthly temperature and precipitation records, with the aim to represent annual trends or seasonality over the 1970–2000 time period (Supplementary Fig. 16; Supplementary Table 5). This analysis randomly samples subsets of sites (encoded as SNP frequencies) that can be partitioned into groups based on differences in climactic variables to estimate the predictive ability of climatic variables. Annual precipitation (BIO12), and temperature annual range (BIO7) were revealed to be the most important bioclimatic variables impacting genetic variation (Fig. 6c).

To identify genes that might be impacted by these variables, we performed a genome-wide test for association between SNP variants, and each of these two environmental variables, accounting for genetic structure between isolates. In total, 17 and 25 significant associations (5% FDR; 49,370 SNPs tested) were found with BIO7 and BIO12, respectively (Fig. 6a bottom panel; Supplementary Table 6). Consistent with the initial differentiation scan, chromosomes I and III harboured most of the associations ($n = 13$ and $n = 11$, respectively). On chromosome I, eight (of the 13) associations with BIO12 fell within a 6 Kbp window (7,177,538 bp and 7,183,617 bp), overlapping the region universally differentiated between climatic areas. Chromosome III contained 11 significant associations for both BIO7 ($n = 6$) and BIO12 ($n = 5$). The two most significant associations were also found on this chromosome for BIO7 (positions 26,824,240 and 26,824,197 bp; LFMM test, $P = 6.34 \times 10^{-9}$ and $P = 1.39 \times 10^{-7}$, respectively). These two SNPs fell within the HCOI00198200 locus, which codes for a metallopeptidase M1, an ortholog of the C. elegans gene anp-1. Additional BIO7-associated SNPs were found within (16,544,889 bp) and in the vicinity (15,905,141 and 15,905,191 bp) of a solute carrier coding gene (HCOI00312500) on chromosome IV, whose orthologs in C. elegans are under the control of daf-12, a key player in dauer formation. Of note, two H. contortus orthologs of components of the dauer pathway in C. elegans, namely tax-4 (HCOI00661100) and daf-36 (HCOI00015800), were among genes under diversifying selection (Supplementary Fig. 14), but neither of these genes contained SNPs identified as being associated with the two most significant bioclimatic variables.

## Discussion

The ecology and epidemiology of gastrointestinal nematodes have been well characterised and exploited to build mathematical models to guide treatment decision in the field[34]. On the contrary, knowledge about their genetic diversity remains limited; a better understanding of population structure and selective pressure applied by environmental factors would yield better predictions of changes in their dispersion[35]. By exploiting a broad collection of samples from globally distributed isolates, together with chromosome-scale assembly and extensive individual resequencing, we have performed an in-depth characterisation of H. contortus genetic diversity, explored historical contributions to its current population structure and identified important drivers shaping the genome of this parasite.

H. contortus populations displayed high levels of nucleotide diversity, consistent with early estimates based on mitochondrial data[36,37], and recent resequencing experiments of inbred isolates[13]. These extreme levels of genetic diversity are thought to arise from both a large census population[36] and the high fecundity of H. contortus females[20].

An early attempt based on a set of genome-wide AFLP markers obtained for 150 individual worms from 14 countries supported the first exploration of H. contortus population structure at the continent level[37]. Analyses of these data identified three to four (Africa, South-East Asia, America and Europe) main phylogenetic clusters as well as evidence for the strong genetic connectivity between Australian, South-African and European isolates[37]. Our genome-wide data corroborated these early results. However, the use of a chromosome-scale assembly and individual resequencing contributed to our ability to identify genome-wide patterns of genetic diversity in its chromosomal context, and in turn, provided sufficient resolution to identify genes likely associated with selective advantage against drug or climate selection pressures. This had not been possible in previous work with AFLP markers[37].

Major past human migrations and associated sheep movements have contributed to the mixing of parasite populations, which partly accounts for the limited genetic structure and extensive admixture between some of our globally distributed isolates. Our data supported an out-of-Africa expansion derived from ancestral populations from Western Africa, with a bottleneck dated back between 2.5 to 10 kya. Sheep domestication originally took place in the Middle East ~ 10 kya[38], before introduction into Eastern Africa and subsequent spread, likely through cultural diffusion, toward southern Africa ~ 2–2.5 kya[39]. The timing of the bottleneck identified in our data and occurring between 2.5 and 10 kya is compatible with major migrations of pastoralist populations that ultimately resulted in the import of small ruminants into Southern Africa[39]. The simultaneous increase in rainfall in central Africa ~ 10.5 kya[40] would have supported the population expansion and dispersal of H. contortus. In addition, early radiation towards Asia observed in our data was congruent with evidence obtained from the study of retroviral insertions within the sheep genome that suggested direct migration between Africa and southwest Asia[41], and with the timing of Asian sheep expansion between 1.8 and 14 kya[42].

The genetic congruence of parasites in Guadeloupe with those from both West Africa and the Mediterranean region is consistent with co-transportation of livestock, including sheep, during the discovery and subsequent colonisation of America. Woolly Churra sheep were originally brought to the Caribbean by Spanish conquistadors[43], before West African breeds more suited to tropical conditions were transported, resulting in an admixed Caribbean sheep population[43]. The timing of genetic admixture here overlaps with centuries of human and presumably livestock movement during the transport of slaves, most of whom originated in West Africa and were transported to colonies that included the French West Indies[21]. The Mediterranean ancestry of H. contortus isolates in Guadeloupe and the close relationship with French worm populations suggest additional sheep transport occurred between French mainland and its Guadeloupian overseas territory. Although these movements are difficult to track

precisely, live sheep were shipped aboard slave trade vessels departing from French harbours[44], and may have introduced European *H. contortus* to the island. Based on historical data[43], a Spanish lineage of *H. contortus* on Guadeloupe would also be expected to exist, though data are missing to confirm this. It can be speculated that the shared ancestry between Guadeloupian and Moroccan worm isolates might result from the introduction of Spanish Churra sheep, as this breed emerged in Spain while the region was under Arab influence between the 8th and 13th century[43,45]. Introduction of *H. contortus*-infected sheep from the Maghreb may have been associated with the spreading of African worms in Spain, and ultimately, Guadeloupe.

The widespread use of anthelmintic drugs has been a major selective force shaping standing genetic variation in the isolates analysed. Adaptation to drug exposure was clearly illustrated in the genetic diversity surrounding the β-tubulin-coding gene, a target of benzimidazole drugs; strong loss of diversity was observed at this locus in many distinct isolates phenotypically characterised to be resistant to this class of drug. The co-occurrence of the same resistant haplotypes in geographically disconnected isolates is almost certainly owing to the independent evolution of benzimidazole resistance as has been described previously[46]; however, some evidence of shared ancestry between French and Guadeloupe resistant individuals emphasises the risk of spreading resistance without careful monitoring of parasite populations during livestock trade. Furthermore, this highlights the risk of evolution of resistance in other parasitic nematodes, for example, parasites of medical interest that are treated with benzimidazoles in mass-drug administration programmes[8].

Ivermectin is an important anthelmintic for parasite control in both veterinary and medical settings[47]. Worldwide emergence of ivermectin-resistant veterinary parasitic nematodes[6] and evidence of reduced efficacy in human filarial nematodes[9] underline the urgency and importance of a better understanding of the mechanisms involved[47]. Our data identified genes previously associated with ivermectin resistance in parasitic nematodes of either veterinary (*pgp-11* in *Parascaris sp*[27].) or medical (*aex-3* in *Onchocerca volvulus*[48]) interest and uncovered candidates. Additional validation will be needed to determine whether these could serve as useful markers in the field to monitor drug efficacy. Of note, support for a major QTL for ivermectin resistance on chromosome V recently identified in a back-cross experiment in two resistant isolates[49] was also present in our genome-wide data and warrants further investigation.

Our broad sampling throughout the global range of *H. contortus* has enabled analyses into climate-driven adaptation in a parasitic nematode. Understanding how climate shapes parasite genetic variation is of primary importance to foresee consequences of climate change on parasite phenology and range dispersion. Parasite dispersal is largely driven by their hosts but *H. contortus* free-living stages experience climatic conditions that affect their development[2] and constrain their spatio-temporal dispersal[50]. Observations in Northern Europe suggest that climate change has already altered *H. contortus* winter phenology[50]. Our analyses identified associations between genetic variation and bioclimatic variables on the one hand, and estimated genome-wide genetic differentiation between populations experiencing contrasting climates. Neither of these two analyses are ideal as bioclimatic variables are correlated and genetic differentiation is influenced by confounding factors such as population ancestry. Nevertheless, the correlation between bioclimatic variables was expected to lead to our analysis underestimating the respective contribution of the most important features[51], but BIO12 and BIO7 were consistently ranked as the most important features. Genetic

differentiation was conserved across various comparisons, suggesting the reported signal was relatively independent of population ancestry. Our results suggest that adaptation toward annual precipitation was mostly under the control of variation on chromosome I, however, no obvious candidate genes could be identified. In addition, the *Hco-tbb-iso-1* locus was close to the region of interest, and linked variation in allele frequency at this locus as a result of benzimidazole selection cannot be ruled out. A second region of chromosome III was associated with both temperature- and precipitation-related variables, within which biologically relevant genes could be identified; first, the strongest genetic associations with annual temperature range were found in a metallopeptidase with zinc ion binding function (*anp-1* ortholog), an enzyme linked to drought stress tolerance in *Drosophila*[52], and second, an ortholog of *Pc* displayed strong genetic differentiation between arid- and wetter temperate or tropical environmental conditions. This gene has been linked to putative epigenetic regulation of xeric adaptation in *Drosophila melanogaster*, where *Pc* mutants display lower resistance to desiccation stress[53]. This finding is also corroborated by observations in plants showing Polycomb-mediated regulation of climate-induced phenotypes[54]. While in the absence of direct experimental evidence we cannot be definitive, we believe it is possible that this function of this *Pc* ortholog is conserved in nematodes. Domain prediction from its peptide sequence indicates a chromo- or chromo-shadow domain, supporting a role in chromatin-binding. In addition, the putative *C. elegans* homologue of this gene[55], named *cec-1*, encodes a protein that recognises the repressive Polycomb-specific H3K27me methylation mark[56]. Knocking out the expression of this gene through RNA interference should help to support its role in the plastic response to climatic stress. Further links to climatic adaptation include identification members of the dauer pathway;[57] *tax-4* and *daf-36* orthologs were under diversifying selection and a *daf-12*-respondent solute carrier was associated with annual temperature range. Dauer is a developmentally arrested stage in *C. elegans* that is triggered by environmental stress and mediates tolerance to unfavourable conditions until better conditions are met[57], and can occur in parasitic nematodes like *H. contortus* under semi-arid conditions[58]. Evidence for climatic adaptation suggests that adaptation in the face of climate change will be constrained by available genetic variability at temperature-selected loci, that may both limit or enable range expansion depending on the region. By a better understanding of the interaction between climatic conditions and phenotypes such as hypobiosis (a temporary developmental arrest during unfavourable conditions), optimisation of treatment timing may be possible to maximise control efficacy.

In summary, our data describe the extensive global and genome-wide diversity of the blood-feeding parasitic nematode *H. contortus*, and how this diversity has been shaped by adaptation to its environment and to drug exposure. Understanding the mechanism(s) by which parasites adapt to fluctuating environmental conditions both within and outside their hosts will have important implications for field management of parasitic nematodes in both veterinary and medical settings. Further characterisation of these putative strategies, together with genetic covariation of drug resistance genes, should contribute to refining epidemiological models and guide treatment decisions for more sustainable management of worm populations in the face of a changing climate.

## Methods

**Sample DNA extraction and sequencing**. A total of 267 individual male *H. contortus* were obtained from a collection held at INRA[17] (metadata for all samples

is presented in detail in Supplementary Data 1). The sampling regime was designed to delineate the contribution of major evolutionary forces, i.e., migration and selection (drug and climate) but also constrained by the material available in the collection. Because migration was likely to match human history, isolates from western African countries and southern America were selected to address the contribution of slave trade history to the structuring of *H. contortus* populations; isolates from former colonies of the British Empire (South-Africa, Australia) were sampled to establish the connectivity between worm populations from Europe and these countries. Ivermectin-resistant isolates (AUS.1, STA.1, STA.3) were also retained to evaluate how anthelmintics had shaped *H. contortus* genomic variability (drug efficacy data have been provided in Supplementary Table 2). Isolates were selected based on available material to ensure minimal sample size ($n = 9$) per isolate and proper allele frequency estimation. Following these criteria, 19 isolates from 12 countries were available (Supplementary Data 1). Note that the second Australian population (AUS.2) was obtained from an Italian laboratory (labelled ITA_NAP, Supplementary Data 1). Samples were gathered between 1995 and 2011 (Supplementary Data 1), and stored in liquid nitrogen upon collection. Four isolates were fenbendazole susceptible (Fig. 1 and Supplementary Table 2; triangles; FRA.2, FRA.4, STO, ZAI).

DNA was extracted with the NucleoSpin Tissue XS kit (Macherey-Nagel GmbH&Co, France) following the manufacturer's instruction. Sequencing libraries were prepared using an amplification-free protocol[13] and were sequenced with 125 bp paired-end reads on an Illumina Hiseq2500 platform using V4 chemistry (Supplementary Data 1). A second round of sequencing was performed with 75 bp paired-end reads to increase the coverage of 43 samples (Supplementary Note 1). After sequencing, two samples were identified to be heavily contaminated by kraken-0.10.6-a2d113dc8f[59] and were discarded. In total, 18 sequencing lanes consisting of 4,152,170,256 reads were sequenced, the raw data of which are archived under the ENA study accession PRJEB9837.

**Sequencing data processing**. Read mapping to both the mitochondrial and nuclear genomes (v3.0, available at ftp://ngs.sanger.ac.uk/production/pathogens/Haemonchus_contortus) was performed using SMALT (http://www.sanger.ac.uk/science/tools/smalt-0) with a median insert size of 500 bp, k-mer length of 13 bp, and a stringency of 90%. For samples that had two or more BAM files (when split across multiple sequencing lanes), the BAM files were merged using samtools v.0.1.19-44428[60], and duplicated reads removed using Picard v.2_14_0 (https://github.com/broadinstitute/picard) before performing realignment around indels using Genome Analysis Toolkit (GATK v3.6)[61] RealignerTargetCreator. Mean coverage of the mitochondrial and genomic genomes were estimated using GATK DepthOfCoverage, revealing coverage lower than the estimated target coverage (original 8×) for most samples. Individuals with >80% of their mitochondrial sequence with at least 15 reads and a mean mitochondrial genome coverage of at least 20× were retained for population genetic inferences ($n = 223$ individuals).

**Nuclear genome SNP calling**. To call SNPs, we used GATK HaplotypeCaller in GVCF mode, followed by joint genotyping across samples (GenotypeGVCFs) and extraction of variants (SelectVariants), resulting in a total of 30,040,159 unfiltered SNPs across the five autosomes. Sex determination in *H. contortus* is based on an XX/XO system; as only male worms were sequenced, their hemizygous X chromosome would have revealed limited phylogenetic information relative to the autosomes, and was henceforth excluded from further analysis.

Low coverage sequencing will inadvertently bias allele sampling at heterozygous SNPs, resulting in excess homozygous genotypes particularly if stringent filtering is applied during SNP calling. To circumvent this issue, we applied the GATK Variant Quality Score Recalibration (https://gatkforums.broadinstitute.org/gatk/discussion/39/variant-quality-score-recalibration-vqsr), which first uses a reference (truth) SNP set to estimate the covariance between called SNP quality score annotations and SNP probabilities, followed by application of these probabilities to the raw SNPs of interest (Supplementary Fig. 17).

The reference "truth" SNP database was generated from the intersection of variants called from samples with at least a mean of 10× coverage ($n = 13$) using three independent SNP callers as described in the dedicated Supplementary Methods section. This procedure yielded 23,868,644 SNPs spanning the five autosomes. Called SNPs were used for particular analyses (*Ne* trajectory through time, cross-population composite likelihood-ratio) that could not be performed under the probabilistic framework that relied on GLs as implemented in ANGSD[62] v. 0.919-20-gb988fab (Supplementary Note 1, Supplementary Fig. 18). These data were also used to estimate average differentiation between isolates (Supplementary Note 1). However, within-isolate diversity and admixture were analysed using ANGSD[62] (Supplementary Note 1).

**Mitochondrial DNA data processing**. The mitochondrial genome exhibited an average coverage depth of 322× (ranging from 24× to 5,868×) per sample (Supplementary Data 1). Mitochondrial reads were extracted and filtered from poorly mapped reads using samtools view (-q 20 -f 0 × 0002 -F 0 × 0004 -F 0 × 0008) and from duplicated reads using Picard v.2_14_0 MarkDuplicates (https://github.com/broadinstitute/picard). Realignment around indels was applied with GATK[61] and SNP were subsequently called using samtools[60] mpileup using only reads that

achieved a mapping quality of 30 and base quality of 30. To exclude sites prone to heterozygous signals from further phylogenetic inference analysis, a SNP calling procedure was implemented with the HaplotypeCaller tool to apply hard filtering parameters on the raw SNP sets (QD ≥ 10, FS ≤ 35 MQ ≥ 30 MQRankSum ≥ −12.5 and ReadPosRankSum ≥ −8) with a minimum depth of 20 reads. This procedure excluded 1354 putative heterozygous sites, and retained 72% of the putative SNP sites (3052 out of 4234 SNPs). Nucleotide diversity and Tajima's D were computed by sliding windows of 100 bp using vcftools v.0.1.15[63]. A PCA was performed on genotypes using the SNPrelate package[64] in R version 3.5[65]. A consensus fasta sequence was subsequently generated with GATK FastaAlternateReferenceMaker for each sample using the filtered variant set, which was used for the phylogenetic analyses.

**Diversity and divergence analysis**. Genome-wide nucleotide diversity ($\pi$) was computed for each isolate with at least five individuals using ANGSD[62]. Using GLs from samtools[60] (option GL = 1) as an input, variants were included that had a minimal supporting evidence of five reads, and base and mapping quality phred scores of at least 20. As $\pi$ values were biased by population mean coverage (Supplementary note 1, Supplementary Fig. 19), $\pi$ was also calculated from a subset of isolates containing individuals with a minimum mean coverage of 5× : this was limited to France (FRA.1, $n = 5$, mean coverage of 7.66×), Guadeloupe ($n = 5$, mean coverage of 12.75×) and Namibia ($n = 6$, mean coverage of 9.85×). A slight increase in Tajima's D values was observed in the same samples following resequencing, i.e., increase in depth of coverage. However, only modest and negative correlations were found between window coverage and Tajima's D, suggesting that coverage is not causing a systematic bias in estimates of Tajima's D (Supplementary Fig. 20).

$F_{ST}$ was estimated from the VQSR-called genotypes between isolates with at least five individuals using the Weir–Cockerham estimator in vcftools v0.1.15[63]. To prevent artefactual signal linked to variation in coverage between isolates, $F_{ST}$ was calculated on subsets of SNPs, binned based on their MAF in 10% increments. The maximum $F_{ST}$ value calculated was retained for comparison (Supplementary Fig. 4). The resulting $F_{ST}$ estimates were not biased by coverage, as measured by negligible correlation between pairwise $F_{ST}$ coefficients and associated population cross-coverage (Pearson's $r_{(136)} = -0.05$, $P = 0.55$; Supplementary Fig. 21).

The pairwise sequence divergence between individual samples was calculated using the Hamming distance, i.e., 1-IBS, using PLINK[66], considering the only individuals with a mean coverage above 2.5× as failure to do so yielded biased estimates (Supplementary Note 1, Supplementary Fig. 22). A neighbour-joining tree of Hamming distances calculated from the nuclear DNA genotypes was built using the R package *ape*[67].

PCA on genotypes inferred from GLs of the 223 samples was generated using ANGSD ngsCovar, filtering for sites with base and mapping quality phred scores < 30, minimum depth of five reads, and a SNP *p* value (as computed by ANGSD) below $10^{-3}$. Clustering was robust to coverage variation and closely matched the PCA from the VQSR-called SNP genotypes (Supplementary Fig. 23).

**Phylogenetic inference**. To determine the phylogenetic structure of the cohort, the 223 consensus mitochondrial fasta sequences were first aligned using Muscle v3.8.31[68], followed by stringent trimming of sequence alignments using Gblocks[69]. The most likely evolutionary model, GTR substitution model with rate heterogeneity modelled by a gamma distribution with invariable sites, was determined using modelgenerator v.0.851[70]. A maximum likelihood tree was subsequently generated using PhyML[71] v.20120412, with branch supports computed using 100 bootstraps.

**Admixture analysis**. Admixture was determined using NGSAdmix[72]. This tool relies on GLs to account for data uncertainty, and has been shown to produce robust inferences about population ancestry from low coverage samples alone, or a mixture of low and higher coverage samples[72]. This analysis was performed on 223 samples, for *K* ranging from 2 to 10 clusters (Supplementary Figs. 5 and 6), retaining sites with <50% missing data across individuals and MAF above 5%. Five iterations were run omitting one autosome out at a time, and the best K was chosen as the first value that would minimise the median absolute deviation across runs (Supplementary Fig. 6). Sample coverage did not affect the results (Supplementary Fig. 24).

**Effective population size and population divergence dating**. The effective population size (*Ne*) trajectory through time, and the cross-divergence time between populations, were estimated using MSMC2[73] as described in the Supplementary Methods section. This approach uses patterns of heterozygosity along the genome to identify past recombination events modelled as Markov processes. Mutation density along the sequence mirrors either recent (long tract of limited diversity) or older (enrichment in heterozygosity over short distances) events. MSMC2 was applied to individuals with a mean coverage above 10×, limiting the analysis to six isolates (FRA.1, GUA, IND, MOR, NAM, STA.1), and considering four haplotypes per isolate. Beagle v4.1 was used to impute missing genotypes and to establish phase in VQSR SNP calls (Supplementary Note 2). Imputation

accuracy analysis revealed a 7.2% and 9.0% discordance rates at the individual and site levels, respectively (Supplementary Fig. 25).

Migration scenarios between populations were determined using the forward simulation framework implemented in $\delta a \delta i$[74] as outlined in the Supplementary Methods section. Under a given evolutionary scenario, this software models the expected joint site frequency spectrum between multiple populations using a diffusion equation. These expected values are then used to compute the most likely demographic parameters knowing the observed site frequency spectrum.

Additional support to the estimates from the nuclear genome were obtained from a phylogenetic analysis of mitochondrial coding sequences as described under the Supplementary Methods section.

**Diversifying selection scan**. To further characterise the genetic diversity in *H. contortus* populations, we identified genomic regions under diversifying selection using XP-CLR[25]. This approach takes advantage of both within-population distortion of the allele frequency spectrum, and between-population differences in allele frequencies in the vicinity of selective sweeps[25]. Although XP-CLR is robust to SNP ascertainment bias[25], the analysis was restricted to isolates with at least five individuals with a minimum mean coverage of 2× per individual. Unphased VQSR-derived genotypes were filtered to retain SNPs with a within-isolate call rate > 80% and MAF > 5%. The analysis was run on every one of the 22 possible within continent pairwise comparisons of the retained isolates (Australia and Africa) with the following options: −w1 0.0001 500 2000 –p0 0. This fit a grid of putatively selected points every 2 Kbp along the genome, with a sliding window size around grid points of 0.01 cM, interpolating SNP position in Morgans from the average recombination rate for each chromosome[13]. Downsampling was applied to windows where > 500 SNPs were found to keep SNP numbers comparable between regions, and no LD-based down-weighing of the CLR scores was applied. A selection score was subsequently computed at every position as the root mean square of XP-CLR coefficients, and the highest 0.1% of selection scores were deemed significant.

**Genetic analysis of benzimidazole- and ivermectin resistance**. The genomic coordinates of *Hco-btub-iso-1* were determined by blasting the gene coding sequence from WormBase Parasite[75] against current genome assembly (BLASTN, *e* value < $10^{-50}$). The SNP positions associated with codons 167, 198 and 200 was determined to be at 7027535, 7027755 and 7027758 bp, respectively, along chromosome I after aligning the *Hco-btub-iso-1* consensus sequence and published sequence (GenBank accession FJ981629.1) using muscle v3.8.31[68]. Genotype and GLs at these positions were determined with ANGSD[62]. Genotypes were only considered for GL > 60% (*n* = 74), as coverage bias occurred otherwise (Supplementary Table 4 and Supplementary Fig. 12). For these samples, phased and imputed genotypes from VQSR SNPs spanning the *Hco-btub-iso-1* locus were used to compute pairwise number of allele differences. Selection in the vicinity of the *Hco-btub-iso-1* locus was assessed by computing Tajima's *D* with ANGSD[62]. For benzimidazole-resistant isolates, neutral state was built with *ms*[76] by simulating 10-Kbp wide isolate-specific sequences (*n* = 1000) following the same coalescent scenario as predicted by MSMC2 for chromosome I (considering a recombination rate of 1.83 cM/Mbp[13]). Suitable *ms* input parameters were derived from MSMC2 output files using the msmc2ms.py script from msmc-tools. This approach was implemented for isolates with sufficient mean depth of coverage, i.e., three benzimidazole-resistant isolates. In case of susceptible isolates, the lack of significant departure in the *Hco-btub-iso-1* vicinity relative to the rest of chromosome I was tested.

The introduction of resistant haplotypes from France mainland into Guadeloupian worm isolates was tested by a topology weighting analysis implemented with TWISST[77]. This method computes phylogenetic trees between a set of isolates, using genetic information from short sliding windows (50 Kbp) and by sampling with replacement individuals from each isolate. At each window, a weight is subsequently computed for each of the possible tree topology, ultimately providing inference of gene flow events in discrete locations where tree topology connects phylogenetically distant isolates. Analyses were run using individuals with sufficient mean depth of coverage (5×and more) from French (FRA.1; *n* = 5; mean depth = 7.62×) and Guadeloupian (GUA; *n* = 5; mean depth = 12.75×) isolates, adding Namibian isolate (*n* = 2; mean depth = 10.9×) as an outgroup. A fourth isolate was chosen for its common ancestry with Guadeloupian isolates: first, an analysis was run with worms from São Tomé (*n* = 2; mean depth = 7.01×), followed by a second analysis using Moroccan samples (*n* = 2; mean depth = 10.6×), to ensure that evidence of gene flow was consistent in both cases.

To investigate the genetic architecture of ivermectin resistance, a differentiation scan was run between ivermectin-resistant isolates and isolates of unknown status sharing same ancestry, i.e., STA.1 and STA.3 vs. NAM and ZAI in Africa, AUS.1 against AUS.2 in Australia. Windowed $F_{ST}$ estimates were calculated every 10 Kbp with a 1 Kbp overlap along the genome with ANGSD[62], retaining sites with minimal depth of five reads, mapping and base quality phred scores of at least 30, missing rate below 50%, and windows with at least 1000 sites. Genomic coordinates of the top 0.5% most-differentiated windows were extracted and analysed by BLASTN (minimum *e* value = $10^{-50}$) against the published V1 *H. contortus*

assembly[14], and annotated gene identifiers were inferred from the corresponding GFF file from WormBase Parasite[75].

**Identification of SNPs under environmental selection**. To identify SNPs putatively influenced by environmental selection, we first defined the climatic conditions of each isolate using the Köppen–Geiger classification inferred from isolate geographical coordinates. Three isolates with the best coverage, i.e., Namibia, France, and Guadeloupe, were used to contrast dry, temperate and tropical conditions, respectively. We performed a genome-wide scan using pairwise comparisons of $F_{ST}$ using ANGSD[62], of which test values >0.5% quantile in at least two comparisons were analysed further.

To further explore the contribution of environmental climatic variables on standing genetic variation, we applied a machine-learning gradient forest algorithm[78] to quantify changes in genetic variation (fit as population SNP MAF) along environmental gradients. The gradients consisted of 19 bioclimatic variables[79], summarising rainfall and temperature information recorded between 1970 and 2000 (Supplementary Table 5). For each SNP, a random forest of 500 trees was grown. For each tree, bootstrapped SNP MAFs were regressed against a random subset of bioclimatic variables to determine the variable that best partitioned the data, thereby building the first node that partitions the data into two sets of homogeneous observations. Iterations follow to determine subsequent nodes by resampling a random subset of bioclimatic variables until no observations are left. The proportion of variance explained by each bioclimatic variable is then averaged across SNPs, and the function of SNP frequency modification along bioclimatic variable is built.

The analysis was performed on isolates with at least five individuals, a mean coverage of 2×, and no record of ivermectin resistance (AUS.2, FRA.1, GUA, IND, MOR, NAM, STO, ZAI). SNP MAFs were estimated from genotype likelihoods with ANGSD, and subsequently filtered to ensure a within-isolate MAF > 10%, at least 90% within-isolate call rate, and shared across the eight considered isolates, resulting in 3758 SNPs retained for further analysis.

Environmental variables were highly correlated (Supplementary Fig. 16a), resulting in instability in the predictor's importance. To minimise this effect, the gradient forest analysis was restricted to the 11 environmental variables showing least redundancy as assessed by a PCA (Supplementary Fig. 16b). Pair-wise Euclidean distances between variables was computed from their respective coordinates on the first two PCA axes (Supplementary Fig. 16a). We selected variables with higher distances (mean distance > 0.85) with any others (BIO4, 7, 2, 15, 5, 10). Other variables defined three clusters (Supplementary Fig. 16b). Within each cluster, we picked the variable with closest distance from every other in the cluster, i.e., the variable summarising others' contributions (BIO9, BIO13, BIO19). For the BIO12, 13, 16, 18 cluster, we chose to pick BIO12 (annual precipitation) which is more relevant toward parasite life cycle across climatic areas. Under temperate areas, quarter-based (BIO16, BIO18) or wettest month (BIO13) statistics would match seasons where hosts are housed. Further variable selection following hierarchical average clustering yielded similar outcomes, suggesting remaining correlations between variables were not affecting the random forest algorithm (Supplementary Fig. 16c). SNP-environment associations were further investigated by focusing on the top two environmental predictors of genomic variations and using a Latent Factor Mixed Model analysis. This analysis was implemented with the lfmm R package[80] on genotype calls from the VQSR pipeline, considering SNPs with call rate of at least 70% across isolates and minor genotype frequency above 5%, using *K* = 3 for the latent factor accounting for underlying population structure. Analyses were run five times and *P* values were combined and adjusted as recommended[80].

**Gene identification and GO enrichment analysis**. Annotation of the revised *H. contortus* genome is ongoing. Therefore, genes underlying major differentiation signals were inferred by searching for similarity between the region of interest (10 Kbp window for $F_{ST}$ analyses, 2000 bp window around XP-CLR hits) against the published V1 *H. contortus* genome assembly[14] using BLAST v2.2.25. Any genes falling within most probable blast hit coordinates were retrieved from the *H. contortus* published assembly available from WormBase Parasite[75]. This database was also used to retrieve *C. elegans* orthologs of positional candidate genes and *H. contortus* Gene Ontology (GO) terms. GO term enrichment analysis was run with the R topGO[81] package, considering nodes with at least 10 annotated genes. The "weight01" algorithm was used to account for existing topology between GO terms. This framework makes *P* value of one GO term conditioned on its neighbours, thereby correcting for multiple testing. Enrichment was tested by the Kolmogorov–Smirnov statistic applied to gene selection or $F_{ST}$ score accordingly. GO terms with *P* values below 1% were deemed significant.

**Reporting summary**. Further information on research design is available in the Nature Research Reporting Summary linked to this article.

## Data availability

Raw sequencing data are archived under the ENA study accession PRJEB9837. Data for figure reproduction have been made available at https://github.com/guiSalle/

Haemonchus_diversity. Reference assembly used in this project is available at: ftp://ngs.sanger.ac.uk/production/pathogens/Haemonchus_contortus.

## Code availability

Sequencing data were analysed with publicly available script and software as mentioned in main text. Outputs were analysed using an R script available at: https://github.com/guiSalle/Haemonchus_diversity.

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

## Acknowledgements

J.A.C., M.B., N.H. and S.D. are supported by the Wellcome Trust via their core funding of the Wellcome Trust Sanger Institute (grant 206194) and by BBSRC grant BB/M003949/1. G.S. has received the support of the EU in the framework of the Marie-Curie FP7 COFUND People Programme, through the award of an AgreenSkills (grant agreement no. 267196) and AgreenSkills + fellowships (grant agreement no. 609398). We are grateful to Robin Beech and John Gilleard for insightful discussions. Nematode material was provided by the BRC4Env Animal Parasitic Nematodes collection, the environmental resources pillar of the AgroBRC-RARe research infrastructure.

## Author contributions

G.S., J.A.C. designed the experiment. G.S., J.A.C., S.D. drafted the manuscript. J.Ca. and J.Co. sampled and prepared parasite materials. G.S. performed DNA extraction and data analyses. S.D. built the reference genome. M.B. and N.H. managed and supervised parasite sequencing and the project.

## Competing interests

The authors declare no competing interests.
