## [Peer Review File · Nature Communications]

Reviewers' Comments:

Reviewer #1:

Remarks to the Author:

This MS investigate the genome-wide diversity of a highly pathogenic parasitic nematode of small ruminants; *Haemonchus contortus*. Unlike in earlier publications this study is based on whole genome by Illumina sequencing based on data from 267 single-worm from 19 populations spanning five continents. This in itself makes this contribution interesting and novel in approach. Also data collection and the analyses are sound and meet high standards.

My major concern is that there is a previous study on the global population genetic structure of *Haemonchus contortus*, in which the genetic variability of the same parasite was assessed by nad4 sequences of the mitochondrial genome and AFLP data (Troell et al IJP 2006). AFLP is also a whole genome method, but for some strange reason Troell et al (2006) is not referred to in the present MS. To me this is surprising, especially as Troell et al. (2006), arrive at a somewhat different conclusions. Characteristic of good science is that the results are discussed in the light of existing knowledge. In my opinion, this is not done in this case. Fortunately, this can be easily corrected simply by relating the new findings to Troell et al. (2006). Also the genes that are putatively linked to ivermectin resistance and provide genetic signatures of climate-driven adaptation could be better highlighted. Overall, I am of the opinion that MS cannot be published until these small corrections are taken into account.

Reviewer #2:

Remarks to the Author:

The authors describe genetic diversity in populations of the parasite *H. contortus*. For the benefits of web searching it may be useful to mention the species name in the title.

The authors undertake a mammoth task of sequencing 223 individuals from 19 populations. However, it was disappointing to discover that many analysis were influenced by low coverage and instead conducted with a limited subset of the data, 3 populations and 16 individuals in the case of nucleotide diversity.

The authors discuss connectivity between populations of *H. contortus* and correlate these events to movement of humans and livestock. However, this did not include any models or testing of alternative hypotheses of divergence times.

The scan for locally adapted alleles was put in the context of climate adaptation. The authors used a variety of techniques to correlate climate variables with highly differentiated alleles. This provided a discussion that could be shaped around climate change. While this is an important discussion, it did not seem convincing for the spread of *H. contortus* as the authors pointed out that Human and livestock are the main transporters. The authors may also want to revisit the statement about "epigenetic regulation" as they provide no data on epigenetics. It was unclear as to how the authors combined samples from 1995 through 2011 with climate data variation.

Major Comments:

-Why was diversity data compared only to *Drosophila melanogaster* and human? There are 2 studies of filarial nematodes that calculate nucleotide diversity, it would seem that those would be better comparisons. (Small 2016, Choi 2016)

-It did not seem as if linkage was taken into account in the calculation of the PCA or Admixture plot. Linkage especially as they identify large selective sweeps could confound tests of population structure.

-PCA/phylogeny/admixture are dependent on ancestry and may not represent the effects of gene flow. Thus it may be more informative to test models of divergence and colonization. This could be done using ABC or dadi.

-how were you able to reach the conclusion in line 190? Lower effective population sizes on islands could lead to more drift or smaller founding populations

-Many of the internal node have no blue dot assuming BS support less than 50%. It would be better to display nodes with BS over 85% or 90% or justify the low cut-off. We are not certain how any conclusion can be drawn from a mtDNA tree with such low support

-It would be more robust to test the migration scenarios rather than infer from the genetic connectivity, The complex interplay of population size, serial founder events, and human movement may confound a simple explanation of the FST and Admixture data.

-There doesn't seem to be data/analysis that backs up the claim on line 227 about genetic connectivity being consistent with slave trade since the slave trade was a temporal event and the authors do not estimate a time here. This is similar for line 231, there is no estimation of the timing of this event.

-we would argue that TajD is not the most robust test for natural selection given that it is influence by demography. One way to remedy this would be to simulate the distribution of TajD along the genome using the reconstructed demography from MSMC. Can the authors also comment on the influence of population mean coverage on these tests since it was found that nucleotide diversity was affected by coverage.

-line 272, Aust and African populations have the same level of divergence at this locus as found genome wide in both nuclear and mitochondrial genomes. That would seem to be evidence for no selection rather than independent origins. It is not clear from Supp Fig10 that Aust and Africa are different.

-Following the above we assumed that Figure 3b was going to demonstrate the haplotype structure in regards to Aust and Africa, but instead was discussing haplotype sharing between FRA and FRG.

-the surrounding windows "congruent with overall population structure" does this mean topo1 and topo2?

-It is also not evident that the highlighted window is unique as downstream there can be seen windows with equally high counts of the topology. We would argue that Twisst which is simply a count of all times that topology is observed in that window by subsampling individuals, is not a robust method to infer selection in recently diverged populations with large Ne. Twisst, in our understanding, is designed to be used to look at introgression between species, which is where the term introgression is usually applied. We do not attempt to argue semantics but maybe gene flow or admixture is more appropriate in this context

Minor Comments:

-Could the authors comment on micro-habitats or discuss how the parasites may experience fine-scale habitat heterogeneity.

-please provide reference as to why XP-CLR is robust to SNP uncertainty

-The authors comment in Figure4b of removing variables that are co-linear, but wouldn't Max Temp or warmest month and Temp annual range be correlated? What was the cutoff for removing correlated factors?

-line 528, backgrounds independently could be due to shared ancestry and standing variation.

-The wrong caption is used for Figure 3C. As it is labeled FST and seems to use all Chromosomes. Also the chromosomes are not labeled in this figure. This appear incorrectly in the body but not the figures appearing at the end.

-limit use of "most". Examples: most pathogenic, most exhaustive genomic survey. Quantitative information would be more useful here. Line 587

-enormously successful and evolutionary successful doesn't really mean anything. How is success measured? Here (we assume) by species diversity, but this could have been a mere 1% of all species,

with 99% going extinct. Evolution is not about success but survival.

-line 83, what is GIN?

-please fix comma splices

-Line 105, not sure what they mean here by "markers", does this mean sites could not be called or SNPs were not segregating in all populations?

-MSMC also can utilize a masking bed file to correctly infer the distribution of heterozygosity. Was this done? Also what parameters as far as rho/mu were used?

-using the same colors in Fig1C for different populations makes it difficult to pick out unique topologies

-what is the analysis to determine step-wise? There was no isolation by distance, so this would be dependent on allele freq and population sizes.

-where do the authors establish that Africa is the origin?

-the authors seem to switch between population names and abbreviations. This made it hard to follow the results as the switching between FRG and Guadeloupien was not obvious.

-Figure3b would be more clear if the authors stacked the distributions rather than overlap

Reviewer #3:

Remarks to the Author:

Sallé et al. The global diversity of a major parasitic nematode is shaped by human intervention and climate

This carefully written paper focusses on the genomic variability of *Haemonchus contortus* from across the globe. The analyses are extensive and detailed, and the results have implications for understanding anthelmintic resistance. The paper is easy to follow and the English is excellent.

This reviewer can find no major flaws in the interpretation of results. The paper will be of general interest, and of more specific interest to medical, veterinary, and parasitology fields.

In general, the first half of the results, dealing with the genetic variability and possible migration patterns, is not as interesting or novel as the results on variability associated with drug resistance and adaptation to climate. That is, the first half contains results that mainly confirm what is expected from the sample size, distance between sampling sites, and known history of human movement, and the results mainly 'confirm' expectations. Therefore, this reviewer would recommend shortening the sections dealing with genetic variation and migration, and save the space for more details about the much more important results in the following sections.

This reviewer has several concerns about the sampling design and definition of 'population':

Line 652: Considering the distribution of this parasite, and of the sampling area, 267 individuals is not a large sample. 'Taking advantage of an available collection' is not motivation for a sampling regime. Please provide more details on the choice of these individuals and how they are deemed to cover most (or which part) of genomic variability.

Line 654: The authors claim that 19 populations were sampled, but what is the definition of population here? Perhaps 'sampling site' would be sufficient?

Line 656: Metadata is said to be detailed in Supplementary Table 1, but this table only provides country names for most 'populations', while the 'populations' in France are equal to farm. Do we assume that each of the other 'populations' in other countries is equivalent to one farm? More details should be provided about the sample origins in Suppl. Table 1, otherwise it is difficult to tell what kind

of 'populations' are being compared. If samples from each country come from smaller (farms) or larger areas (we cannot tell for most countries), this could change the interpretation of the results.

Reviewers' comments:

Reviewer #1 (Remarks to the Author):

This MS investigate the genome-wide diversity of a highly pathogenic parasitic nematode of small ruminants; *Haemonchus contortus*. Unlike in earlier publications this study is based on whole genome by Illumina sequencing based on data from 267 single-worm from 19 populations spanning five continents. This in itself makes this contribution interesting and novel in approach. Also data collection and the analyses are sound and meet high standards.

My major concern is that there is a previous study on the global population genetic structure of *Haemonchus contortus*, in which the genetic variability of the same parasite was assessed by *nad4* sequences of the mitochondrial genome and AFLP data (Troell et al IJP 2006). AFLP is also a whole genome method, but for some strange reason Troell et al (2006) is not referred to in the present MS. To me this is surprising, especially as Troell et al. (2006), arrive at a somewhat different conclusions. Characteristic of good science is that the results are discussed in the light of existing knowledge. In my opinion, this is not done in this case. Fortunately, this can be easily corrected simply by relating the new findings to Troell et al. (2006).

We agree that the Troell et al. study should have been cited and discussed.

The study by Troell et al. considered a sample size of 150 worms from 14 countries. They made use of 1429 AFLP markers and the *nad4* mitochondrial gene sequence. AFLP-based results were more informative and could better resolve worm population structure than the *nad4* mitochondrial gene. Nonetheless, our results are fairly congruent with the study by Troell et al. They reported mitochondrial-based average nucleotide diversity ($\text{Pi} = 0.037$ [0.001;0.03] for the *nad4* sequence) in the range of our values [0.004;0.013] based on whole mitogenome. This reference had been cited in our discussion of the observed nucleotide diversity levels: “*H. contortus* populations displayed high levels of nucleotide diversity, consistent with early estimates based on mitochondrial data^{39,40}, and recent re-sequencing experiments of inbred isolates¹²”.

They also found similar population structure at the continent level and identified three to four (Africa, South-East Asia, America and Europe) main phylogenetic clusters. Of note, they also identified the strong genetic connectivity between Australian, South-African and European isolates. To this regard, we don't feel that our results differ greatly.

Last, our SNP information combine to a chromosome-scale assembly offers greater resolution by both covering every genomic region and pointing at genes of interest. We could also establish the pattern of genomic variation in their chromosomal context which was not possible in the study by Troell et al.

Mention to this previous work and the advantages offered by our data have been inserted under the discussion section as follows:

” *An early attempt based on a set of genome-wide AFLP markers obtained for 150 individual worms from 14 countries supported the first exploration of H. contortus population structure at the continent level³⁹. Analyses of these data identified three to four (Africa, South-East Asia, America and Europe) main phylogenetic clusters as well as evidence for the strong genetic connectivity between Australian, South-African and European isolates³⁹. Our genome wide data corroborated these early results. However, the use of a chromosome-scale assembly and individual resequencing contributed to identify genome-wide patterns of genetic diversity in its chromosomal context, and in turn, provided sufficient resolution to identify genes likely associated with selective advantage against drug or climate selection pressures. This had not been possible in previous attempts with AFLP markers³⁹.*”

Also the genes that are putatively linked to ivermectin resistance and provide genetic signatures of climate-driven adaptation could be better highlighted. Overall, I am of the opinion that MS cannot be published until these small corrections are taken into account.

To address this comment, we have updated the results section where relevant, with a focus on the most convincing candidates for drug resistance phenotype and climate adaptation.

- Genetic diversity estimates in the vicinity of the *Pc* ortholog and a gene-wise differentiation scan with base-pair resolution has been performed, revealing likely mutations in exon 5 of this gene associated with amino-acid changes in each of the 3 considered populations (FRA.1, FRG and NAM) for which individuals with higher mean depth of coverage are available.
- For ivermectin resistance, individuals with coverage depth were not available and identification of causative mutations is therefore not possible. Instead nucleotide diversity between Australian isolates has been provided for the three candidates for ivermectin resistance identified across all XP-CLR comparisons (orthologs of *pgp-11* and *glc-4*). These two isolates share similar ancestry, have similar mean depth of coverage and validated ivermectin resistance status. In that case, our results favour higher diversity in the ivermectin-resistant isolate.
- Same plots were generated for genes highlighted in XP-CLR tests with the STA.1 isolate (orthologs of *aex-3*, *snf-9*, *unc-24* and *B0361.4*). In that case, the STA.1 isolate displayed reduced nucleotide diversity relative to other African populations.
- We also released a supplementary table (supplementary table 6) with the complete list of candidates sharing overlap with XP-CLR hits that would benefit the community for further functional investigations that are beyond the scope of our manuscript.

Reviewer #2 (Remarks to the Author):

The authors describe genetic diversity in populations of the parasite *H. contortus*. For the benefits of web searching it may be useful to mention the species name in the title.

Title has been modified accordingly.

The authors undertake a mammoth task of sequencing 223 individuals from 19 populations. However, it was disappointing to discover that many analyses were influenced by low coverage and instead conducted with a limited subset of the data, 3 populations and 16 individuals in the case of nucleotide diversity.

We, of course, share the reviewers disappointment with the low coverage we obtained from our sequencing, and that the resulting dataset was difficult to handle. Despite this, we have put a great deal of effort into cautious evaluation on how coverage was affecting the results and have implemented up-to-date frameworks to deal with this issue. We hope the reviewer agree that it is better to be conservative in these analyses than to 'push the data' too far.

The authors discuss connectivity between populations of *H. contortus* and correlate these events to movement of humans and livestock. However, this did not include any models or testing of alternative hypotheses of divergence times.

Estimation of divergence times have now been performed and reported and simulations used to assess the relative strength of these conclusions.

The scan for locally adapted alleles was put in the context of climate adaptation. The authors used a variety of techniques to correlate climate variables with highly differentiated alleles. This provided a discussion that could be shaped around climate change. While this is an important discussion, it did not seem convincing for the spread of *H. contortus* as the authors pointed out that Human and livestock are the main transporters.

We agree, and neither our discussion nor results argue that climate acts as a migratory force. Instead, we claim that this environmental constraint imposes selection and contributes to shaping standing genetic variation.

The authors may also want to revisit the statement about "epigenetic regulation" as they provide no data on epigenetics.

We agree with the referee's view and this statement has been modified as: “*revealing that a gene acting as an epigenetic regulator*”.

It was unclear as to how the authors combined samples from 1995 through 2011 with climate data variation.

Our analysis regarding climate adaptation is mostly centred on genetic differentiation between three populations. In that case, FRA and NAM were collected at a 1-year interval but FRG was sampled more recently. Because the genetic differentiation signal holds across every possible comparison, we believe the temporal effect is not at play.

Additional analyses tackling adaptation to climatic conditions were run for 8 isolates sampled between 1995 and 2005. A temporal disconnection is involved between MOR, FRG, AUS.2 on one hand and FRA.1, IND, NAM, STO and ZAI on the other hand. Nevertheless, bioclimatic data are mean values aggregating records of a 30-year time period between 1970 and 2000 (now added under the results and materials and methods sections). Our analysis hence establishes how standing genetic variation at the time of sampling was shaped by climatic conditions associated with different geographical regions experienced during this time.

Major Comments:

- Why was diversity data compared only to *Drosophila melanogaster* and human? There are 2 studies of filarial nematodes that calculate nucleotide diversity, it would seem that those would be better comparisons. (Small 2016, Choi 2016)

Human and *D. melanogaster* nucleotide diversities were originally chosen as lower and upper bounds. Nucleotide diversity values for suggested species have been added to Figure S1, and main text and Figure S1 caption were amended accordingly.

- It did not seem as if linkage was taken into account in the calculation of the PCA or Admixture plot. Linkage especially as they identify large selective sweeps could confound tests of population structure.

Linkage disequilibrium was not considered for pruning our SNP set, mainly because LD inference from low-coverage data is not trivial. We agree this should be done, and have now done so:

To overcome this issue, ANGSD was run as previously mentioned to estimate minor allele frequencies at filtered sites for admixture and PCA estimation. These MAFs were further used to compute pairwise linkage disequilibrium (r^2) using the recently published maximum likelihood framework by Bilton et al. Genetics 2018. Sites were subsequently pruned to retain marker pairs showing $r^2 < 0.2$. Figures have been updated to show results after sites editing. Both admixture and PCA results were only marginally affected by this procedure, although explained variation along first two PCA axes was slightly reduced to 2.86 and 1.67%.

Similarly, PCA based on VQSR SNP set (supplementary Figure 21) was recomputed using the same workflow as previously mentioned but applying a threshold of 0.2 for LD, hence retaining 13,893 SNPs out of the 411,574 SNPs that passed low MAF (<5%) and low call rate (<50%) filters. The resulting pattern was similar to the one obtained without LD filtering, although explained variance was slightly reduced (7.96% and 5.86% for first and 2nd axis respectively).

The Results and the Materials & Methods section have been updated accordingly.

- PCA/phylogeny/admixture are dependent on ancestry and may not represent the effects of gene flow. Thus it may be more informative to test models of divergence and colonization. This could be done using ABC or dadi.

To address this comment, we have performed simulations using dadi (see our response regarding migration scenario testing and dating below).

- how were you able to reach the conclusion in line 190? Lower effective population sizes on islands could lead to more drift or smaller founding populations

Insular populations usually suffer higher exposure to drift or derive from smaller founding populations. Both factors are expected to reduce within population genetic diversity which in turn inflates F_{ST} estimates (Charlesworth 1998, *Mol. Biol. Evol.* 15: 538–543). We hence assumed that higher F_{ST} observed for this population was reflecting their insular situation and likely exposure to higher drift. Inferences from joint demography (dadi) between STO and other populations suggest isolation occurred in the XVIIth century, despite a more complex scenario with MOR population.

To address this comment, we have revised the wording in current manuscript:

“Inferences of the joint history of STO and other African populations supported this view with different scenarios of ancient migration but isolation since 1560 at least (Table 1). These inferences suggested a secondary contact with MOR population took place in late 1800’s (Table 1) which was also evident by the admixture pattern between STO and MOR (Fig. 2c).”

- Many of the internal node have no blue dot assuming BS support less than 50%. It would be better to display nodes with BS over 85% or 90% or justify the low cut-off. We are not certain how any conclusion can be drawn from a mtDNA tree with such low support.

We acknowledge this issue and interpret it as an insufficient number of polymorphisms to fully delineate the relationships between admixed populations, combined with a fairly deep tree. This issue certainly has been exacerbated by the conservative nature of the bootstrapping algorithm which expects strictly identical topologies across replicates and does not account for partially matching topologies between replicates (issue addressed in Lemoine et al. 2018 *Nature* 556, 452-456).

To improve tree stability, we produced an unrooted tree that slightly improved branch support values (mean increasing from 54% to 58%). We have also updated the tree representation, blue dots now showing support of at least 70% following Hillis et al. 1993 *Syst. Biol.* 42, 182-192. A few deep nodes among the haplogroup of Mediterranean ancestry and one of the São Tomé/Guadeloupean clade still had low support. Nevertheless, this does not hamper our main conclusion about population clustering. The materials and methods section has been updated accordingly to describe our analysis.

- It would be more robust to test the migration scenarios rather than infer from the genetic connectivity, The complex interplay of population size, serial founder events, and human movement may confound a simple explanation of the F_{ST} and Admixture data.

- There doesn't seem to be data/analysis that backs up the claim on line 227 about genetic connectivity being consistent with slave trade since the slave trade was a temporal event and the authors do not estimate a time here. This is similar for line 231, there is no estimation of the timing of this event.

We agree, and have now implemented tests of these ideas using dadi. Specifically, forward simulations from joint site frequency spectra have been implemented with dadi using available scripts (https://github.com/dportik/dadi_pipeline) encoding 4 rounds of estimation with 10, 20 30 and 40 replicates respectively. 2D-SFS was estimated with ANGSD and we implemented a hierarchical model selection strategy by first comparing Akaike Criterion (AIC) of scenarios with no migration against that of simple (a)symmetrical migrations. When migration history was more likely, more complex models were subsequently implemented and ranked according to their AIC (the less, the better). The sole exception to this framework was for FRA.1-NAM and FRA.1-STA.3: in that case

AIC was close between models with and without migration but estimated timings made more sense for more complex models that were hence retained.

Initial exploration of the fit of more complex scenarios than a simple “split and isolation” model suggested there was little power in our data to estimate parameters accurately under these models. We hence reported and discussed timings obtained from simple models that provided support to major migration events, *i.e.* slave trade and colonization of Australia. Nevertheless, outputs from the more complex models were discussed as an indication of the most likely demographic scenario between corresponding populations. The full list of parameter estimates has been provided in supplementary table 3. The methods section was also updated as follows:

“Most likely migratory scenarios between populations were determined using the forward simulation framework implemented in `dadi`⁹⁰. Under a given evolutionary scenario, this software models the expected joint site frequency spectrum between multiple populations using a diffusion equation. These expected values are then used to compute the most likely demographic parameters knowing the observed site frequency spectrum. For each model, four rounds of forward simulations with 10, 20, 30 and 40 replicates respectively using previously published python scripts⁹¹. Model Akaike Information Criterion (AIC) were compared for ranking, the less, the better.

We first compared a scenario without migration against simple symmetrical and asymmetrical gene flow. In case migration was the most likely, more complex models (involving split with (a)symmetrical gene flow, with or without population size change, or models involving secondary contact with/without gene flow) were tested. However, initial exploration indicated a lack of power in our design to accurately estimate parameters of more complex demographic models than the “split and isolation” model. Nevertheless, these models still provide the most likely scenario and their output have been listed in supplementary Table 3.

Parameters were scaled to real time using same parameters as for MSMC2 inference. Standard deviations of timing estimates for the simple split and isolation models were obtained using the Godambe Information Matrix⁹⁵ applied to 100 simulated site frequency spectra produced with the `ms` software under the most likely demographic model.“

- we would argue that TajD is not the most robust test for natural selection given that it is influence by demography. One way to remedy this would be to simulate the distribution of TajD along the genome using the reconstructed demography from MSMC.

Windowed Tajima’s *D* was estimated throughout the genome. We agree upon the fact that Tajima’s *D* suffers sensitivity to population demography, but would argue that the ‘background’ genome-wide estimates would account for this demography to provide an empirical null distribution from which the particular locus under scrutiny depart. However, we agree that it is worth doing this in a more rigorous way, and as suggested have used the MSMC inferred demography from genome wide data to simulate chromosome segments using the `ms` software to get Tajima’s *D* estimates.

For the two considered susceptible populations (AUS.2 and ZAI) however, mean of coverage was too low to have robust MSMC inference estimated. In that case, we did not run the MSMC software and took the genome-wide Tajima’s *D* estimate as a mean reference level.

The materials & methods section (***Analysis of β -tubulin isotype 1 (Hco-btub-iso-1) and the genetic architecture of benzimidazole- and ivermectin-resistance***) has been updated to describe `ms` simulations as follows:

*“Selection in the vicinity of the Hco-btub-iso-1 locus was assessed by computing Tajima’s *D* with ANGSD91. For benzimidazole-resistant populations, neutral state was built with `ms92` by simulating a 1,000 10-Kbp wide population-specific sequences following the same coalescent scenario as predicted by MSMC for chromosome I (considering a recombination rate of 1.83 cM/Mbp¹²). Suitable `ms` input parameters were derived from MSMC output files using the `msmc2ms.py` script from `msmc-tools`. This*

approach was implemented for populations with sufficient mean depth of coverage, i.e. three benzimidazole-resistant populations. In case of susceptible populations, the lack of significant departure in the Hco-tub-iso-1 vicinity relative to the rest of chromosome I was tested.”

Results now show simulated Tajima's D expectations for resistant populations where selection is thought to have occurred. Figure 3a caption was also updated.

- Can the authors also comment on the influence of population mean coverage on these tests since it was found that nucleotide diversity was affected by coverage.

We are not sure about which tests the reviewer is referring to and assume this is Tajima's *D*. A short mention on the impact of coverage for Tajima's *D* estimation had been provided in supplementary Figure 17. We found that Tajima's *D* was slightly higher with higher coverage, but chromosome-wide patterns were similar.

- line 272, Aust and African populations have the same level of divergence at this locus as found genome wide in both nuclear and mitochondrial genomes. That would seem to be evidence for no selection rather than independent origins. It is not clear from Supp Fig10 that Aust and Africa are different.

- Following the above we assumed that Figure 3b was going to demonstrate the haplotype structure in regards to Aust and Africa, but instead was discussing haplotype sharing between FRA and FRG.

Our point was to establish the similarity between FRA and GUA populations at this locus that is congruent with gene flow. Discussion about Australian and African populations has been removed.

- the surrounding windows “congruent with overall population structure” does this mean topo1 and topo2?

Yes, this is what was meant. We have added topology names to avoid confusion.

- It is also not evident that the highlighted window is unique as downstream there can be seen windows with equally high counts of the topology. We would argue that Twisst which is simply a count of all times that topology is observed in that window by subsampling individuals, is not a robust method to infer selection in recently diverged populations with large N_e . Twisst, in our understanding, is designed to be used to look at introgression between species, which is where the term introgression is usually applied. We do not attempt to argue semantics but maybe gene flow or admixture is more appropriate in this context.

We agree that gene flow would be a more appropriate terminology than introgression and we have edited the main text accordingly. Nonetheless, our point was not to use Twisst to detect selection but to emphasize existing genetic connectivity at the β -tubulin locus. This approach seems to confirm that gene flow has been involved between FRA.1 and GUA populations over this locus. Surrounding region also seem to share equivalent pattern which could arise from hitchhiking after selection. To better illustrate our view, we now provide plots spanning a wider genomic window (100 Kbp). We also inverted Fig3b and supplementary Fig11 as the results appears to be clearer with a topology using the Moroccan population.

Minor Comments:

- Could the authors comment on micro-habitats or discuss how the parasites may experience fine-scale habitat heterogeneity.

Yes, and the following comment has been inserted into the discussion section: “*Parasite dispersal is largely driven by their hosts but H. contortus free-living stages experience climatic conditions that affect their development and constrain their spatio-temporal dispersal. Observations in Northern Europe suggest that climate change has already altered H. contortus winter phenology.*”

- please provide reference as to why XP-CLR is robust to SNP uncertainty

XP-CLR robustness to SNP data has been demonstrated in the original XP-CLR paper. Reference was available under the Materials & methods section and has been added under the results section.

- The authors comment in Figure 4b of removing variables that are co-linear, but wouldn't Max Temp or warmest month and Temp annual range be correlated? What was the cutoff for removing correlated factors?

Factors were selected to limit redundancy between variables. To do so, we computed their pair-wise Euclidean distances from their respective coordinates on the first two PCA components (supplementary Fig. 16a). We selected variables with higher distances (mean distance > 0.85) to any others (bio4, 7, 2, 15, 5, 10). Other variables formed three clusters of closely correlated variables. Within each cluster, we chose the variable closest to the centroid of the cluster, i.e. the variable summarizing others' contributions (bio9, bio13, bio19), however for one cluster (containing bio12, 13, 16, 18) we chose to pick bio12 (annual precipitation) as likely to be more relevant to parasite life-cycles across climatic areas. This strategy has been described further under the materials and methods section, but we note that our results are – as would be expected – appear to be robust to precisely which variate within correlated clusters was used: for example, picking bio13 instead yielded the same variable classification.

- line 528, backgrounds independently could be due to shared ancestry and standing variation.

To address this comment, we have rephrased this line to:

“The co-occurrence of resistant haplotypes in disconnected isolates could result from selection acting upon variants appeared in several backgrounds independently⁵¹, or from shared ancestry as found between France and Guadeloupe resistant individuals. This latter observation”

-The wrong caption is used for Figure 3C. As it is labeled FST and seems to use all Chromosomes. Also the chromosomes are not labeled in this figure. This appear incorrectly in the body but not the figures appearing at the end.

Figure caption has been modified accordingly.

- limit use of “most”. Examples: most pathogenic, most exhaustive genomic survey. Quantitative information would be more useful here. Line 587

We agree. and have limited the use of “most” throughout the text, including those the reviewer has pointed to, e.g. “*Most exhaustive survey*” has been replaced with “*This exhaustive survey of genome-wide diversity*” and most pathogenic was rephrased as “*haematophagous*”.

- enormously successful and evolutionary successful doesn't really mean anything. How is success measured? Here (we assume) by species diversity, but this could have been a mere 1% of all species, with 99% going extinct. Evolution is not about success but survival.

This statement has been discarded.

- line 83, what is GIN?

GIN stands for Gastro-Intestinal Nematode and is now written explicitly.

- pleas fix comma splices

We have modified the text accordingly.

- Line 105, not sure what they mean here by “markers”, does this mean sites could not be called or SNPs were not segregating in all populations?

We assessed how many of the retained SNPs were shared across populations. We found 411,574 SNPs with MAF>5% and call rate of at least 50%. Without MAF restriction, *i.e.* including non-segregating SNPs, 3,338,155 were called in 50% of available individuals.

To address this comment, the sentence has been rephrased as follows:

“Only a limited fraction of the filtered SNPs se markers was called in more than half the individuals (n = 3,338,155 SNPs) and shared across populations (n = 411,574 SNPs) were segregating across common across more than half the individuals with MAF>5%.”

- MSMC also can utilize a masking bed file to correctly infer the distribution of heterozygosity. Was this done? Also what parameters as far as rho/mu were used?

Masking bed file and mappability files were indeed created as recommended by MSMC authors. The corresponding materials and methods section (*Effective population size and population divergence dating*) has been updated accordingly:

“Input files were created following MSMC recommendations and available msmc-tools (<https://github.com/stschiff/msmc-tools/blob/master/README.md>). Briefly, the reference fasta sequence was masked with SNPable (<http://lh3lh3.users.sourceforge.net/snpable.shtml>) to extract regions of unambiguous read mapping in chromosome-specific bed files (using the msmc_create_map_mask.py python script). Negative bed files indicating regions with sufficient coverage at the individual level were created from samples bam files with the bamCaller.py script and filter out sites with coverage below genome-wide average depth. Finally, MSMC input files were created for each chromosome with the generate_multihetsep.py script and concatenated into a single input file.”

Our original analysis was run using the default rho/mu value (0.25). Following referee’s suggestion, we have updated our analyses using rho = 1.68 cM/Mbp (Doyle et al. GBE 2018) and assuming a mutation rate equal to that reported for *C. elegans* (2.7×10^{-9} per site per generation), resulting in a rho/mu value of $1.68\% / 10^6 / 2.7 \times 10^{-9} = 6.22$. Results and conclusions remained similar.

The materials and methods section have been updated accordingly:

*“For each population, estimates were averaged across five runs, leaving one chromosome out at a time for cross-validation and using rho/mu parameter value of 6.22 (average recombination rate of 1.68 cM/Mbp¹² and assuming a mutation rate similar to that of *C. elegans* mutation rate, 2.7×10^{-9} per site per generation²⁰).”*

- using the same colors in Fig1C for different populations makes it difficult to pick out unique topologies

To account for this comment, we have redesigned Fig1c with different colors to better highlight considered populations.

- what is the analysis to determine step-wise? There was no isolation by distance, so this would be dependent on allele freq and population sizes.

This statement has been withdrawn and the whole section updated to account for dadi simulation output.

- where do the authors establish that Africa is the origin?

We established that African populations are more diverse than other populations and that a bottleneck occurred in non-African populations around 10 Kya (MSMC inference). Forward simulations also indicated isolation between European and African populations occurred somewhat earlier. It seems therefore likely that non-African populations represent a subset of original diversity, after migration out of Africa.

- the authors seem to switch between population names and abbreviations. This made it hard to follow the results as the switching between FRG and Guadeloupian was not obvious.

To address this comment, FRG was renamed as GUA to ease the reading.

- Figure3b would be more clear if the authors stacked the distributions rather than overlap

Figure 3b and corresponding supplementary figure 11 have been modified accordingly.

Reviewer #3 (Remarks to the Author):

Sallé et al. The global diversity of a major parasitic nematode is shaped by human intervention and climate

This carefully written paper focusses on the genomic variability of *Haemonchus contortus* from across the globe. The analyses are extensive and detailed, and the results have implications for understanding anthelmintic resistance. The paper is easy to follow and the English is excellent.

This reviewer can find no major flaws in the interpretation of results. The paper will be of general interest, and of more specific interest to medical, veterinary, and parasitology fields.

We thank the reviewer for their appraisal of our work.

In general, the first half of the results, dealing with the genetic variability and possible migration patterns, is not as interesting or novel as the results on variability associated with drug resistance and adaptation to climate. That is, the first half contains results that mainly confirm what is expected from the sample size, distance between sampling sites, and known history of human movement, and the results mainly 'confirm' expectations. Therefore, this reviewer would recommend shortening the sections dealing with genetic variation and migration, and save the space for more details about the much more important results in the following sections.

Following the first reviewer's comments, we have better highlighted the genes of interest. However, the questions raised by the second referee made it impossible to shorten sections about migration. Additional simulation analyses have been performed that brought relevant insights and significant modifications to our previous findings and understanding of *H. contortus* history. To this regard and to fit the requested concision in the formatting, we stuck to the original weight given to these sections.

This reviewer has several concerns about the sampling design and definition of 'population':

Line 652: Considering the distribution of this parasite, and of the sampling area, 267 individuals is not a large sample. 'Taking advantage of an available collection' is not motivation for a sampling regime.

Please provide more details on the choice of these individuals and how they are deemed to cover most (or which part) of genomic variability.

The sampling regime was motivated to delineate the contribution of major evolutionary forces, *i.e.* migration and selection (drug, climate). Because migration was likely to match human history, isolates from western African countries and West Indies were selected to address the contribution of slave trade history to *H.contortus* populations, and former colonies of the British Empire (South-Africa, Australia) were sampled to establish the connectivity between worm populations from these countries. Ivermectin-resistant isolates were also retained to evaluate how anthelmintics had shaped *H.contortus* genomic variability. Isolates were selected based on available material to ensure minimal sample size (n = 9) per isolate for proper allele frequency estimation. Following these criteria, populations from 12 countries were available. We accept that the available samples do not represent an ideal sample collection, but we are constrained by the availability of parasite material from other regions.

To address this comment, we have expanded on the description of sampling in the materials and methods section.

Line 654: The authors claim that 19 populations were sampled, but what is the definition of population here? Perhaps ‘sampling site’ would be sufficient?

Population refers to field isolate, collected from one sampling site (farm) in each case.

To address this remark, this definition has been brought to the manuscript under the materials and methods section. The term “*population*” was replaced with “*isolate*” throughout the manuscript.

Line 656: Metadata is said to be detailed in Supplementary Table 1, but this table only provides country names for most ‘populations’, while the ‘populations’ in France are equal to farm. Do we assume that each of the other ‘populations’ in other countries is equivalent to one farm? More details should be provided about the sample origins in Suppl. Table 1, otherwise it is difficult to tell what kind of ‘populations’ are being compared. If samples from each country come from smaller (farms) or larger areas (we cannot tell for most countries), this could change the interpretation of the results.

Each isolate was sampled from one farm unless stated otherwise.

This precision has been brought to the supplementary table legend.

Reviewers' Comments:

Reviewer #2:

Remarks to the Author:

The author's did a commendable job addressing our comments. We still require some clarifications to the below comments.

Key:

R2: Reviewer 2

A: Author response

R2A: Reviewer 2 response

--

R2: how were you able to reach the conclusion in line 190? Lower effective population sizes on islands could lead to more drift or smaller founding populations

A: Insular populations usually suffer higher exposure to drift or derive from smaller founding populations. Both factors are expected to reduce within population genetic diversity which in turn inflates F_{ST} estimates (Charlesworth 1998, *Mol. Biol. Evol.* 15: 538–543). We hence assumed that higher F_{ST} observed for this population was reflecting their insular situation and likely exposure to higher drift. Inferences from joint demography (dadi) between STO and other populations suggest isolation occurred in the XVIIth century, despite a more complex scenario with MOR population.

R2A: To clarify, our comment was aimed at the assumption that high F_{ST} was due to isolation. High F_{ST} , as noted in your response can be due to small founding populations or high drift, likely confounding. However, F_{ST} is a measure of drift and not really a good estimator of migration (Whitlock and McCauley 1999). So we interpreted isolation to mean that you were assuming no migration. Your response is adequate, however, we can not find the quoted statement in the text. We can only locate "Inference of the joint history of STO and other African population supported this view." There is no mention of "secondary contact" in manuscript.

R2: Can the authors also comment on the influence of population mean coverage on these tests since it was found that nucleotide diversity was affected by coverage.

A: We are not sure about which tests the reviewer is referring to and assume this is Tajima's D . A short mention on the impact of coverage for Tajima's D estimation had been provided in supplementary Figure 17. We found that Tajima's D was slightly higher with higher coverage, but chromosome-wide patterns were similar.

R2A: Yes, we were referring to Taj D as your results indicated an affect on estimates of the pairwise diversity (π). If the Taj D value is more positive with higher coverage (possible more accurate estimates of π), will it also be biased to be more negative in lower coverage regions?

R2A: The authors agreed that introgression is not the proper terminology and are said to have replaced it, yet it appears in the manuscript Lines: 621, 1021, 1027, 1035

R2A: The authors said to have changed FRG to GUA, but FRG is still present in much of the manuscript and having FRG and GUA refer to the same thing is quite confusing. Lines 296, 299, Fig 4A, Fig 1A

R2: line 528, backgrounds independently could be due to shared ancestry and standing variation.

A2: To address this comment, we have rephrased this line to:

"The co-occurrence of resistant haplotypes in disconnected isolates could result from selection acting upon variants appeared in several backgrounds independently 51 , or from shared ancestry as found between France and Guadeloupe resistant individuals. This latter observation"

R2A: The sentence does not appear in the manuscript as there is no mention of independent backgrounds. Instead Line 593 reads: "The co-occurrence of the same resistant haplotypes in geographically disconnected isolates is almost certainly due to the independent evolution of benzimidazole resistance as has been described previously". I dont understand how the authors can claim that it is certainly due to independent evolution without testing the hypothesis with their data.

Further comments:

The climate change influencing genetic variation is still problematic since:

- 1) Only 3 comparisons
- 2) No account for shared population history when identifying the loci
- 3) The method of using a PCA to select from highly correlated variables, does not remove the correlation.
- 4) Reference Drosophila for all gene ontologies

Reviewers' comments:

Reviewer #2 (Remarks to the Author):

The author's did a commendable job addressing our comments. We still require some clarifications to the below comments.

Key:

R2: Reviewer 2

A: Author response

R2A: Reviewer 2 response

R2: how were you able to reach the conclusion in line 190? Lower effective population sizes on islands could lead to more drift or smaller founding populations

--

A: Insular populations usually suffer higher exposure to drift or derive from smaller founding populations. Both factors are expected to reduce within population genetic diversity which in turn inflates F_{ST} estimates (Charlesworth 1998, *Mol. Biol. Evol.* 15: 538–543). We hence assumed that higher F_{ST} observed for this population was reflecting their insular situation and likely exposure to higher drift. Inferences from joint demography (dadi) between STO and other populations suggest isolation occurred in the XVIIth century, despite a more complex scenario with MOR population.

--

R2A: To clarify, our comment was aimed at the assumption that high F_{ST} was due to isolation. High F_{ST} , as noted in your response can be due to small founding populations or high drift, likely confounding. However, F_{ST} is a measure of drift and not really a good estimator of migration (Whitlock and McCauley 1999). So we interpreted isolation to mean that you were assuming no migration. Your response is adequate, however, we can not find the quoted statement in the text. We can only locate "Inference of the joint history of STO and other African population supported this view." There is no mention of "secondary contact" in manuscript.

Following latest rewording of the manuscript, it appears the original statement (the one provided in our response) was modified to (L197-203): *"Inference of the joint history of STO and other African populations supported this view. The best-fitting model (supplementary Table 3) was consistent with either ancient symmetrical gene flow followed by isolation (since 1560 at least) or an early split followed by secondary contact in late 1800's before isolation with MOR population (supplementary Table 3). This latter demography would underpin the pattern of admixture detected between STO and MOR (Fig. 2c)."*

We apologize for this discrepancy. To make this statement clearer we inserted "as a result of higher exposure to drift or smaller founding populations" (L198) and added "*in late 1800's*" (L203).

R2: Can the authors also comment on the influence of population mean coverage on these tests since it was found that nucleotide diversity was affected by coverage.

A: We are not sure about which tests the reviewer is referring to and assume this is Tajima's D . A short mention on the impact of coverage for Tajima's D estimation had been provided in supplementary Figure 17. We found that Tajima's D was slightly higher with higher coverage, but chromosome-wide patterns were similar.

R2A: Yes, we were referring to Taj D as your results indicated an affect on estimates of the pairwise diversity (π). If the Taj D value is more positive with higher coverage (possible more accurate estimates of π), will it also be biased to be more negative in lower coverage regions?

We agree that local depth of coverage could bias our analysis but our data indicate that this phenomenon is not occurring. We retrieved the local depth of coverage using *vcftools* for the 23 million SNPs identified and computed a windowed average coverage for the three populations with highest coverage (FRA, GUA, NAM). Correlation between average window coverage and the respective estimated Tajima's D was weak in all cases (Pearson's $r = -0.17, -0.13$ and -0.04 for FRA,

GUA and NAM populations). This new result has been provided in supplementary Figure 20 and the following has been inserted under the materials and methods section (L 919): “A slight increase in Tajima’s D values was observed in the same samples following resequencing, i.e. increase in depth of coverage. However, only modest and negative correlations were found between window coverage and Tajima’s D estimates suggesting that coverage is not causing a systematic bias in estimates of Tajima’s D (supplementary Fig. 20).”

R2A: The authors agreed that introgression is not the proper terminology and are said to have replaced it, yet it appears in the manuscript Lines: 621, 1021, 1027, 1035

These were indeed overlooked. To account for this, the following modifications have been brought:

- L318: “genetic material”
- L635: “back-cross experiment”
- L1066: “The introduction of resistant haplotypes from France mainland into Guadeloupian”
- L1071: “ultimately providing inference of gene flow events in discrete”
- L1080: “to ensure that evidence of gene flow was consistent in both cases.”

Supplementary material was also updated.

R2A: The authors said to have changed FRG to GUA, but FRG is still present in much of the manuscript and having FRG and GUA refer to the same thing is quite confusing. Lines 296, 299, Fig 4A, Fig 1A

We apologize for this issue. This has now been fixed with the appropriate Figure 1A being inserted in main text and corrections have been brought to lines 296 and 299 (now 308 and 311). Supplementary figures 1, 3, 4, 6, 10 and 23 and their respective captions have also been updated.

R2: line 528, backgrounds independently could be due to shared ancestry and standing variation.

A2: To address this comment, we have rephrased this line to: “The co-occurrence of resistant haplotypes in disconnected isolates could result from selection acting upon variants appeared in several backgrounds independently 51 , or from shared ancestry as found between France and Guadeloupe resistant individuals. This latter observation”

R2A: The sentence does not appear in the manuscript as there is no mention of independent backgrounds. Instead Line 593 reads: "The co-occurrence of the same resistant haplotypes in geographically disconnected isolates is almost certainly due to the independent evolution of benzimidazole resistance as has been described previously". I dont understand how the authors can claim that it is certainly due to independent evolution without testing the hypothesis with their data.

To clarify this, we have built a neighbour-net graph based on allelic divergence observed for the 296 SNPs called over the beta-tubulin locus between the 74 samples with sufficient coverage. This network shows resistant populations are polyphyletic, consistent with independent origin of beta-tubulin mutations. This was even true within the French population (FRA.1) that was split into two subclades. However, the clustering of FRA.1 and FRG reflected the already described shared ancestry. The heatmap of allelic differences between samples over the beta-tubulin locus was replaced with the split network to make our point clearer (supplementary Figure 10). A sentence was added to the result section (L295):”A phylogenetic network based on genotype information over the whole *Hco-tbb-iso-1* locus (Supplementary Fig. 10) revealed that resistant populations were polyphyletic suggesting resistant mutations have generally evolved independently in different populations, e.g. Namibia (NAM) and South-Africa (STA.1, STA.3) and to a lesser extent in Australia, however French and Guadeloupian individuals had little divergence in their haplotypes, suggesting that gene flow of resistant haplotypes between mainland France and the West Indies may have occurred (Supplementary Fig. 10).”.

Further comments:

The climate change influencing genetic variation is still problematic since:

1) Only 3 comparisons

We originally focused on these three populations as they had the best coverage. Nevertheless, comparisons between additional populations (with a mean coverage above 3 and at least 5 individuals sequenced) also yield the same differentiation signal on chromosome III (supplementary figure 15). In every case but between Namibia and Indonesia, the genomic windows overlapping the *Pc* ortholog locus were departing from the F_{ST} distribution. These additional results have been provided as a supplementary Figure 15 and commented under the result section (L451): “*Additional comparisons between populations with different histories, i.e. Australia, Indonesia, São Tomé and Zaire also supported this signal (supplementary Figure 15) suggesting it was independent of the precise choice of population included.*”

2) No account for shared population history when identifying the loci

The fact that this differentiation signal holds across a wide range of comparisons performed would favour its relative independence from population ancestry. In addition, differentiation signal was found between populations with common ancestry, *e.g.* Namibia and Zaire or Australia and Indonesia; this suggests that differential ancestry is not confounding this finding.

3) The method of using a PCA to select from highly correlated variables, does not remove the correlation.

We agree that our PCA approach has not removed all of the correlations between the climate variables we model, but do not think this has had a major effect on the conclusions of this work.

Firstly, the PCA approach, identifying a small number of climatic variables representing each highly co-linear group of variables, has removed much of the correlation present.

Secondly, we have used a random forest algorithm that attempts to account for correlation between variables by computing variable importance using a conditional permutation on correlated variables, that is intended to counteract the “correlation bias” inherent in these methods (Tolosi et al. 2011, Bioinformatics). While this “correlation bias” is expected to result in the contribution of groups of correlated variables being downweighed, in fact we see remaining correlations between BIO7, BIO4 and BIO12 – variables that are consistently ranked as the most important features – suggested that correlation bias is having little effect on the limited number of features considered in our analysis.

Finally, we have further investigated this by performing hierarchical average clustering on the variables that remained after our PCA based variable selection to explore remaining correlation structure (an approach followed by Huang et al. 2003, The Lancet (cited in Tolosi et al. 2011)). This approach identified 5 main clusters composed of BIO4, BIO12, BIO19, BIO2 and BIO15, and a last cluster containing BIO7, BIO5, BIO9 and BIO10. We ran a random forest analysis on the subset of variables from these clusters (BIO4, BIO12, BIO19, BIO2 and BIO7) and found BIO12 and BIO7 as the most important features. The same conclusion was found with BIO15 instead of BIO2. Our analysis identifying the importance of BIO12 and BIO7-thus seems robust to the details of variable selection.

These details were added in supplementary Fig. 16c and under the methods section (L1136):” *Further variable selection following hierarchical average clustering yielded similar outcomes, suggesting remaining correlations between variables were not affecting the random forest algorithm (supplementary Fig. 16c).*”

We have also modified the discussion to make it clear that these variables stand for a group of highly correlated climate variables and to recall that none of our approaches is ideal (L. 645): “*Our analyses identified associations between genetic variation and bioclimatic variables on the one hand, and estimated genome-wide genetic differentiation between populations experiencing contrasting climates. Neither of these two analyses are ideal, as bioclimatic variables are correlated and genetic differentiation is influenced by confounding factors such as population ancestry. Nevertheless, the correlation between bioclimatic variables was expected to lead to our analysis underestimating the*

respective contribution of the most important features, but BIO12 and BIO7 were consistently ranked as the most important features. Genetic differentiation was conserved across various comparisons suggesting the reported signal was relatively independent of population ancestry.”

4) Reference *Drosophila* for all gene ontologies

Our gene ontology enrichment analyses were based on the gene ontology terms assigned to gene models in the *H. contortus* v1.0 genome. These terms were assigned using WormBase GO terms via orthology with *C. elegans* and from interproscan protein domains.

Our discussion of *Drosophila* is because there is evidence for the function of our *Pc* ortholog candidate gene in that species. While in the absence of direct experimental evidence we cannot be definitive, we believe it is at least plausible that this function is conserved:

- domain prediction from its peptide sequence indicates a chromo- or chromo shadow domain, supporting a role in chromatin-binding.
- One-to-one orthology between *HCOI01540500* and *Pc* in *Drosophila melanogaster* was originally found in WormBaseParasite. Current data from the WormBaseParasite database (current identifier *HCON_00087060*) suggest that no one-to-one ortholog of this gene was found in *C. elegans* but present in 11 other clade V nematode species including two free-living organisms (*O. tipulae*, *C. angaria*) and 9 parasitic species (*A. ceylanicum*, *A. duodenale*, *D. viviparous*, *H. placei*, *H. polygyrus*, *N. americanus*, *N. brasiliensis*, *O. dendatum*).
- In addition, derived orthologs in mammals (chromobox, *cbx*) and a putative homolog in *C. elegans* (*cec-1*) was reported by Whitcomb et al. Trends in Genetics 2007. More recent functional insights (Saltzman et al. Genetics 2018) seem to support this homology with evidence that the repressive Polycomb-specific H3K27me methylation mark was recognized by the CEC-1 protein (and CEC-6, another chromo domain protein).

Some of these elements have been inserted under the discussion section to emphasise that functional evidence is still missing but supported by other insights (L. 670-677): “*While in the absence of direct experimental evidence we cannot be definitive, we believe it is possible that this function of this Pc ortholog is conserved in nematodes. Domain prediction from its peptide sequence indicates a chromo- or chromo-shadow domain supporting a role in chromatin-binding. In addition, the putative C. elegans homolog of this gene⁵⁹, named cec-1, encodes a protein that recognizes the repressive Polycomb-specific H3K27me methylation mark⁶⁰. Knocking out the expression of this gene through RNA interference should help to support its role in the plastic response to climatic stress.*”

Reviewers' Comments:

Reviewer #2:

Remarks to the Author:

No further comments. Thank you for addressing the additional comments and concerns.

Reviewers' comments:

Reviewer #1 (Remarks to the Author):

This MS investigate the genome-wide diversity of a highly pathogenic parasitic nematode of small ruminants; *Haemonchus contortus*. Unlike in earlier publications this study is based on whole genome by Illumina sequencing based on data from 267 single-worm from 19 populations spanning five continents. This in itself makes this contribution interesting and novel in approach. Also data collection and the analyses are sound and meet high standards.

My major concern is that there is a previous study on the global population genetic structure of *Haemonchus contortus*, in which the genetic variability of the same parasite was assessed by *nad4* sequences of the mitochondrial genome and AFLP data (Troell et al IJP 2006). AFLP is also a whole genome method, but for some strange reason Troell et al (2006) is not referred to in the present MS. To me this is surprising, especially as Troell et al. (2006), arrive at a somewhat different conclusions. Characteristic of good science is that the results are discussed in the light of existing knowledge. In my opinion, this is not done in this case. Fortunately, this can be easily corrected simply by relating the new findings to Troell et al. (2006).

We agree that the Troell et al. study should have been cited and discussed.

The study by Troell et al. considered a sample size of 150 worms from 14 countries. They made use of 1429 AFLP markers and the *nad4* mitochondrial gene sequence. AFLP-based results were more informative and could better resolve worm population structure than the *nad4* mitochondrial gene. Nonetheless, our results are fairly congruent with the study by Troell et al. They reported mitochondrial-based average nucleotide diversity ($\Pi = 0.037$ [0.001;0.03] for the *nad4* sequence) in the range of our values [0.004;0.013] based on whole mitogenome. This reference had been cited in our discussion of the observed nucleotide diversity levels: “*H. contortus* populations displayed high levels of nucleotide diversity, consistent with early estimates based on mitochondrial data^{39,40}, and recent re-sequencing experiments of inbred isolates¹²”.

They also found similar population structure at the continent level and identified three to four (Africa, South-East Asia, America and Europe) main phylogenetic clusters. Of note, they also identified the strong genetic connectivity between Australian, South-African and European isolates. To this regard, we don't feel that our results differ greatly.

Last, our SNP information combine to a chromosome-shape assembly offers greater resolution by both covering every genomic region and pointing at genes of interest. We could also establish the pattern of genomic variation in their chromosomal context which was not possible in the study by Troell et al.

Mention to this previous work and the advantages offered by our data have been inserted under the discussion section as follows:

” *An early attempt based on a set of genome-wide AFLP markers obtained for 150 individual worms from 14 countries supported the first exploration of H. contortus population structure at the continent level³⁹. Analyses of these data identified three to four (Africa, South-East Asia, America and Europe) main phylogenetic clusters as well as evidence for the strong genetic connectivity between Australian, South-African and European isolates³⁹. Our genome wide data corroborated these early results. However, the use of a chromosome-scale assembly and individual resequencing contributed to identify genome-wide patterns of genetic diversity in its chromosomal context, and in turn, provided sufficient resolution to identify genes likely associated with selective advantage against drug or climate selection pressures. This had not been possible in previous attempts with AFLP markers³⁹.*”

Also the genes that are putatively linked to ivermectin resistance and provide genetic signatures of climate-driven adaptation could be better highlighted. Overall, I am of the opinion that MS cannot be published until these small corrections are taken into account.

To address this comment, we have updated the results section where relevant, with a focus on the most convincing candidates for drug resistance phenotype and climate adaptation.

- Genetic diversity estimates in the vicinity of the *Pc* ortholog and a gene-wise differentiation scan with base-pair resolution has been performed, revealing likely mutations in exon 5 of this gene associated with amino-acid changes in each of the 3 considered populations (FRA.1, FRG and NAM) for which individuals with higher mean depth of coverage are available.
- For ivermectin resistance, individuals with coverage depth were not available and identification of causative mutations is therefore not possible. Instead nucleotide diversity between Australian isolates has been provided for the three candidates for ivermectin resistance identified across all XP-CLR comparisons (orthologs of *pgp-11* and *glc-4*). These two isolates share similar ancestry, have similar mean depth of coverage and validated ivermectin resistance status. In that case, our results favour higher diversity in the ivermectin-resistant isolate.
- Same plots were generated for genes highlighted in XP-CLR tests with the STA.1 isolate (orthologs of *aex-3*, *snf-9*, *unc-24* and *B0361.4*). In that case, the STA.1 isolate displayed reduced nucleotide diversity relative to other African populations.
- We also released a supplementary table (supplementary table 6) with the complete list of candidates sharing overlap with XP-CLR hits that would benefit the community for further functional investigations that are beyond the scope of our manuscript.

Reviewer #2 (Remarks to the Author):

The authors describe genetic diversity in populations of the parasite *H. contortus*. For the benefits of web searching it may be useful to mention the species name in the title.

Title has been modified accordingly.

The authors undertake a mammoth task of sequencing 223 individuals from 19 populations. However, it was disappointing to discover that many analyses were influenced by low coverage and instead conducted with a limited subset of the data, 3 populations and 16 individuals in the case of nucleotide diversity.

We, of course, share the reviewers disappointment with the low coverage we obtained from our sequencing, and that the resulting dataset was difficult to handle. Despite this, we have put a great deal of effort into cautious evaluation on how coverage was affecting the results and have implemented up-to-date frameworks to deal with this issue. We hope the reviewer agree that it is better to be conservative in these analyses than to 'push the data' too far.

The authors discuss connectivity between populations of *H. contortus* and correlate these events to movement of humans and livestock. However, this did not include any models or testing of alternative hypotheses of divergence times.

Estimation of divergence times have now been performed and reported and simulations used to assess the relative strength of these conclusions.

The scan for locally adapted alleles was put in the context of climate adaptation. The authors used a variety of techniques to correlate climate variables with highly differentiated alleles. This provided a discussion that could be shaped around climate change. While this is an important discussion, it did not seem convincing for the spread of *H. contortus* as the authors pointed out that Human and livestock are the main transporters.

We agree, and neither our discussion nor results argue that climate acts as a migratory force. Instead, we claim that this environmental constraint imposes selection and contributes to shaping standing genetic variation.

The authors may also want to revisit the statement about "epigenetic regulation" as they provide no data on epigenetics.

We agree with the referee's view and this statement has been modified as: "*revealing that a gene acting as an epigenetic regulator*".

It was unclear as to how the authors combined samples from 1995 through 2011 with climate data variation.

Our analysis regarding climate adaptation is mostly centred on genetic differentiation between three populations. In that case, FRA and NAM were collected at a 1-year interval but FRG was sampled more recently. Because the genetic differentiation signal holds across every possible comparison, we believe the temporal effect is not at play.

Additional analyses tackling adaptation to climatic conditions were run for 8 isolates sampled between 1995 and 2005. A temporal disconnection is involved between MOR, FRG, AUS.2 on one hand and FRA.1, IND, NAM, STO and ZAI on the other hand. Nevertheless, bioclimatic data are mean values aggregating records of a 30-year time period between 1970 and 2000 (now added under the results and materials and methods sections). Our analysis hence establishes how standing genetic variation at the time of sampling was shaped by climatic conditions associated with different geographical regions experienced during this time.

Major Comments:

- Why was diversity data compared only to *Drosophila melanogaster* and human? There are 2 studies of filarial nematodes that calculate nucleotide diversity, it would seem that those would be better comparisons. (Small 2016, Choi 2016)

Human and *D. melanogaster* nucleotide diversities were originally chosen as lower and upper bounds. Nucleotide diversity values for suggested species have been added to Figure S1, and main text and Figure S1 caption were amended accordingly.

- It did not seem as if linkage was taken into account in the calculation of the PCA or Admixture plot. Linkage especially as they identify large selective sweeps could confound tests of population structure.

Linkage disequilibrium was not considered for pruning our SNP set, mainly because LD inference from low-coverage data is not trivial. We agree this should be done, and have now done so:

To overcome this issue, ANGSD was run as previously mentioned to estimate minor allele frequencies at filtered sites for admixture and PCA estimation. These MAFs were further used to compute pairwise linkage disequilibrium (r^2) using the recently published maximum likelihood framework by Bilton et al. Genetics 2018. Sites were subsequently pruned to retain marker pairs showing $r^2 < 0.2$. Figures have been updated to show results after sites editing. Both admixture and PCA results were only marginally affected by this procedure, although explained variation along first two PCA axes was slightly reduced to 2.86 and 1.67%.

Similarly, PCA based on VQSR SNP set (supplementary Figure 21) was recomputed using the same workflow as previously mentioned but applying a threshold of 0.2 for LD, hence retaining 13,893 SNPs out of the 411,574 SNPs that passed low MAF (<5%) and low call rate (<50%) filters. The resulting pattern was similar to the one obtained without LD filtering, although explained variance was slightly reduced (7.96% and 5.86% for first and 2nd axis respectively).

The Results and the Materials & Methods section have been updated accordingly.

- PCA/phylogeny/admixture are dependent on ancestry and may not represent the effects of gene flow. Thus it may be more informative to test models of divergence and colonization. This could be done using ABC or dadi.

To address this comment, we have performed simulations using dadi (see our response regarding migration scenario testing and dating below).

- how were you able to reach the conclusion in line 190? Lower effective population sizes on islands could lead to more drift or smaller founding populations

Insular populations usually suffer higher exposure to drift or derive from smaller founding populations. Both factors are expected to reduce within population genetic diversity which in turn inflates F_{ST} estimates (Charlesworth 1998, *Mol. Biol. Evol.* 15: 538–543). We hence assumed that higher F_{ST} observed for this population was reflecting their insular situation and likely exposure to higher drift. Inferences from joint demography (dadi) between STO and other populations suggest isolation occurred in the XVIIth century, despite a more complex scenario with MOR population.

To address this comment, we have revised the wording in current manuscript:

“Inferences of the joint history of STO and other African populations supported this view with different scenarios of ancient migration but isolation since 1560 at least (Table 1). These inferences suggested a secondary contact with MOR population took place in late 1800’s (Table 1) which was also evident by the admixture pattern between STO and MOR (Fig. 2c).”

- Many of the internal node have no blue dot assuming BS support less than 50%. It would be better to display nodes with BS over 85% or 90% or justify the low cut-off. We are not certain how any conclusion can be drawn from a mtDNA tree with such low support.

We acknowledge this issue and interpret it as an insufficient number of polymorphisms to fully delineate the relationships between admixed populations, combined with a fairly deep tree. This issue certainly has been exacerbated by the conservative nature of the bootstrapping algorithm which expects strictly identical topologies across replicates and does not account for partially matching topologies between replicates (issue addressed in Lemoine et al. 2018 *Nature* 556, 452-456).

To improve tree stability, we produced an unrooted tree that slightly improved branch support values (mean increasing from 54% to 58%). We have also updated the tree representation, blue dots now showing support of at least 70% following Hillis et al. 1993 *Syst. Biol.* 42, 182-192. A few deep nodes among the haplogroup of Mediterranean ancestry and one of the São Tomé/Guadeloupian clade still had low support. Nevertheless, this does not hamper our main conclusion about population clustering. The materials and methods section has been updated accordingly to describe our analysis.

- It would be more robust to test the migration scenarios rather than infer from the genetic connectivity, The complex interplay of population size, serial founder events, and human movement may confound a simple explanation of the F_{ST} and Admixture data.

- There doesn't seem to be data/analysis that backs up the claim on line 227 about genetic connectivity being consistent with slave trade since the slave trade was a temporal event and the authors do not estimate a time here. This is similar for line 231, there is no estimation of the timing of this event.

We agree, and have now implemented tests of these ideas using dadi. Specifically, forward simulations from joint site frequency spectra have been implemented with dadi using available scripts (https://github.com/dportik/dadi_pipeline) encoding 4 rounds of estimation with 10, 20 30 and 40 replicates respectively. 2D-SFS was estimated with ANGSD and we implemented a hierarchical model selection strategy by first comparing Akaike Criterion (AIC) of scenarios with no migration against that of simple (a)symmetrical migrations. When migration history was more likely, more complex models were subsequently implemented and ranked according to their AIC (the less, the better). The sole exception to this framework was for FRA.1-NAM and FRA.1-STA.3: in that case

AIC was close between models with and without migration but estimated timings made more sense for more complex models that were hence retained.

Initial exploration of the fit of more complex scenarios than a simple “split and isolation” model suggested there was little power in our data to estimate parameters accurately under these models. We hence reported and discussed timings obtained from simple models that provided support to major migration events, *i.e.* slave trade and colonization of Australia. Nevertheless, outputs from the more complex models were discussed as an indication of the most likely demographic scenario between corresponding populations. The full list of parameter estimates has been provided in supplementary table 3. The methods section was also updated as follows:

“Most likely migratory scenarios between populations were determined using the forward simulation framework implemented in `dadi`⁹⁰. Under a given evolutionary scenario, this software models the expected joint site frequency spectrum between multiple populations using a diffusion equation. These expected values are then used to compute the most likely demographic parameters knowing the observed site frequency spectrum. For each model, four rounds of forward simulations with 10, 20, 30 and 40 replicates respectively using previously published python scripts⁹¹. Model Akaike Information Criterion (AIC) were compared for ranking, the less, the better.

We first compared a scenario without migration against simple symmetrical and asymmetrical gene flow. In case migration was the most likely, more complex models (involving split with (a)symmetrical gene flow, with or without population size change, or models involving secondary contact with/without gene flow) were tested. However, initial exploration indicated a lack of power in our design to accurately estimate parameters of more complex demographic models than the “split and isolation” model. Nevertheless, these models still provide the most likely scenario and their output have been listed in supplementary Table 3.

Parameters were scaled to real time using same parameters as for MSMC2 inference. Standard deviations of timing estimates for the simple split and isolation models were obtained using the Godambe Information Matrix⁹⁵ applied to 100 simulated site frequency spectra produced with the `ms` software under the most likely demographic model.“

- we would argue that TajD is not the most robust test for natural selection given that it is influence by demography. One way to remedy this would be to simulate the distribution of TajD along the genome using the reconstructed demography from MSMC.

Windowed Tajima’s *D* was estimated throughout the genome. We agree upon the fact that Tajima’s *D* suffers sensitivity to population demography, but would argue that the ‘background’ genome-wide estimates would account for this demography to provide an empirical null distribution from which the particular locus under scrutiny depart. However, we agree that it is worth doing this in a more rigorous way, and as suggested have used the MSMC inferred demography from genome wide data to simulate chromosome segments using the `ms` software to get Tajima’s *D* estimates.

For the two considered susceptible populations (AUS.2 and ZAI) however, mean of coverage was too low to have robust MSMC inference estimated. In that case, we did not run the MSMC software and took the genome-wide Tajima’s *D* estimate as a mean reference level.

The materials & methods section (*Analysis of β -tubulin isotype 1 (Hco-btub-iso-1) and the genetic architecture of benzimidazole- and ivermectin-resistance*) has been updated to describe `ms` simulations as follows:

*“Selection in the vicinity of the Hco-btub-iso-1 locus was assessed by computing Tajima’s *D* with ANGSD91. For benzimidazole-resistant populations, neutral state was built with `ms92` by simulating a 1,000 10-Kbp wide population-specific sequences following the same coalescent scenario as predicted by MSMC for chromosome I (considering a recombination rate of 1.83 cM/Mbp¹²). Suitable `ms` input parameters were derived from MSMC output files using the `msmc2ms.py` script from `msmc-tools`. This*

approach was implemented for populations with sufficient mean depth of coverage, i.e. three benzimidazole-resistant populations. In case of susceptible populations, the lack of significant departure in the Hco-btub-iso-1 vicinity relative to the rest of chromosome I was tested.”

Results now show simulated Tajima's *D* expectations for resistant populations where selection is thought to have occurred. Figure 3a caption was also updated.

- Can the authors also comment on the influence of population mean coverage on these tests since it was found that nucleotide diversity was affected by coverage.

We are not sure about which tests the reviewer is referring to and assume this is Tajima's *D*. A short mention on the impact of coverage for Tajima's *D* estimation had been provided in supplementary Figure 17. We found that Tajima's *D* was slightly higher with higher coverage, but chromosome-wide patterns were similar.

- line 272, Aust and African populations have the same level of divergence at this locus as found genome wide in both nuclear and mitochondrial genomes. That would seem to be evidence for no selection rather than independent origins. It is not clear from Supp Fig10 that Aust and Africa are different.

- Following the above we assumed that Figure 3b was going to demonstrate the haplotype structure in regards to Aust and Africa, but instead was discussing haplotype sharing between FRA and FRG.

Our point was to establish the similarity between FRA and GUA populations at this locus that is congruent with gene flow. Discussion about Australian and African populations has been removed.

- the surrounding windows “congruent with overall population structure” does this mean topo1 and topo2?

Yes, this is what was meant. We have added topology names to avoid confusion.

- It is also not evident that the highlighted window is unique as downstream there can be seen windows with equally high counts of the topology. We would argue that Twisst which is simply a count of all times that topology is observed in that window by subsampling individuals, is not a robust method to infer selection in recently diverged populations with large N_e . Twisst, in our understanding, is designed to be used to look at introgression between species, which is where the term introgression is usually applied. We do not attempt to argue semantics but maybe gene flow or admixture is more appropriate in this context.

We agree that gene flow would be a more appropriate terminology than introgression and we have edited the main text accordingly. Nonetheless, our point was not to use Twisst to detect selection but to emphasize existing genetic connectivity at the β -tubulin locus. This approach seems to confirm that gene flow has been involved between FRA.1 and GUA populations over this locus. Surrounding region also seem to share equivalent pattern which could arise from hitchhiking after selection. To better illustrate our view, we now provide plots spanning a wider genomic window (100 Kbp). We also inverted Fig3b and supplementary Fig11 as the results appears to be clearer with a topology using the Moroccan population.

Minor Comments:

- Could the authors comment on micro-habitats or discuss how the parasites may experience fine-scale habitat heterogeneity.

Yes, and the following comment has been inserted into the discussion section: “*Parasite dispersal is largely driven by their hosts but H. contortus free-living stages experience climatic conditions that affect their development and constrain their spatio-temporal dispersal. Observations in Northern Europe suggest that climate change has already altered H. contortus winter phenology.*”

- please provide reference as to why XP-CLR is robust to SNP uncertainty

XP-CLR robustness to SNP data has been demonstrated in the original XP-CLR paper. Reference was available under the Materials & methods section and has been added under the results section.

- The authors comment in Figure 4b of removing variables that are co-linear, but wouldn't Max Temp or warmest month and Temp annual range be correlated? What was the cutoff for removing correlated factors?

Factors were selected to limit redundancy between variables. To do so, we computed their pair-wise Euclidean distances from their respective coordinates on the first two PCA components (supplementary Fig. 16a). We selected variables with higher distances (mean distance > 0.85) to any others (bio4, 7, 2, 15, 5, 10). Other variables formed three clusters of closely correlated variables. Within each cluster, we chose the variable closest to the centroid of the cluster, i.e. the variable summarizing others' contributions (bio9, bio13, bio19), however for one cluster (containing bio12, 13, 16, 18) we chose to pick bio12 (annual precipitation) as likely to be more relevant to parasite life-cycles across climatic areas. This strategy has been described further under the materials and methods section, but we note that our results are – as would be expected – appear to be robust to precisely which variate within correlated clusters was used: for example, picking bio13 instead yielded the same variable classification.

- line 528, backgrounds independently could be due to shared ancestry and standing variation.

To address this comment, we have rephrased this line to:

“The co-occurrence of resistant haplotypes in disconnected isolates could result from selection acting upon variants appeared in several backgrounds independently⁵¹, or from shared ancestry as found between France and Guadeloupe resistant individuals. This latter observation”

-The wrong caption is used for Figure 3C. As it is labeled FST and seems to use all Chromosomes. Also the chromosomes are not labeled in this figure. This appear incorrectly in the body but not the figures appearing at the end.

Figure caption has been modified accordingly.

- limit use of “most”. Examples: most pathogenic, most exhaustive genomic survey. Quantitative information would be more useful here. Line 587

We agree. and have limited the use of “most” throughout the text, including those the reviewer has pointed to, e.g. “*Most exhaustive survey*” has been replaced with “*This exhaustive survey of genome-wide diversity*” and most pathogenic was rephrased as “*haematophagous*”.

- enormously successful and evolutionary successful doesn't really mean anything. How is success measured? Here (we assume) by species diversity, but this could have been a mere 1% of all species, with 99% going extinct. Evolution is not about success but survival.

This statement has been discarded.

- line 83, what is GIN?

GIN stands for Gastro-Intestinal Nematode and is now written explicitly.

- please fix comma splices

We have modified the text accordingly.

- Line 105, not sure what they mean here by “markers”, does this mean sites could not be called or SNPs were not segregating in all populations?

We assessed how many of the retained SNPs were shared across populations. We found 411,574 SNPs with $MAF > 5\%$ and call rate of at least 50%. Without MAF restriction, *i.e.* including non-segregating SNPs, 3,338,155 were called in 50% of available individuals.

To address this comment, the sentence has been rephrased as follows:

“Only a limited fraction of the filtered SNPs se markers was called in more than half the individuals ($n = 3,338,155$ SNPs) and shared across populations ($n = 411,574$ SNPs were segregating across common across more than half the individuals with $MAF > 5\%$).”

- MSMC also can utilize a masking bed file to correctly infer the distribution of heterozygosity. Was this done? Also what parameters as far as ρ/μ were used?

Masking bed file and mappability files were indeed created as recommended by MSMC authors. The corresponding materials and methods section (*Effective population size and population divergence dating*) has been updated accordingly:

“Input files were created following MSMC recommendations and available msmc-tools (<https://github.com/stschiff/msmc-tools/blob/master/README.md>). Briefly, the reference fasta sequence was masked with SNPable (<http://lh3lh3.users.sourceforge.net/snpable.shtml>) to extract regions of unambiguous read mapping in chromosome-specific bed files (using the `msmc_create_map_mask.py` python script). Negative bed files indicating regions with sufficient coverage at the individual level were created from samples bam files with the `bamCaller.py` script and filter out sites with coverage below genome-wide average depth. Finally, MSMC input files were created for each chromosome with the `generate_multihetsep.py` script and concatenated into a single input file.”

Our original analysis was run using the default ρ/μ value (0.25). Following referee’s suggestion, we have updated our analyses using $\rho = 1.68$ cM/Mbp (Doyle et al. GBE 2018) and assuming a mutation rate equal to that reported for *C. elegans* (2.7×10^{-9} per site per generation), resulting in a ρ/μ value of $1.68\% / 10^6 / 2.7 \times 10^{-9} = 6.22$. Results and conclusions remained similar.

The materials and methods section have been updated accordingly:

*“For each population, estimates were averaged across five runs, leaving one chromosome out at a time for cross-validation and using ρ/μ parameter value of 6.22 (average recombination rate of 1.68 cM/Mbp12 and assuming a mutation rate similar to that of *C. elegans* mutation rate, 2.7×10^{-9} per site per generation20).”*

- using the same colors in Fig1C for different populations makes it difficult to pick out unique topologies

To account for this comment, we have redesigned Fig1c with different colors to better highlight considered populations.

- what is the analysis to determine step-wise? There was no isolation by distance, so this would be dependent on allele freq and population sizes.

This statement has been withdrawn and the whole section updated to account for dadi simulation output.

- where do the authors establish that Africa is the origin?

We established that African populations are more diverse than other populations and that a bottleneck occurred in non-African populations around 10 Kya (MSMC inference). Forward simulations also indicated isolation between European and African populations occurred somewhat earlier. It seems therefore likely that non-African populations represent a subset of original diversity, after migration out of Africa.

- the authors seem to switch between population names and abbreviations. This made it hard to follow the results as the switching between FRG and Guadeloupian was not obvious.

To address this comment, FRG was renamed as GUA to ease the reading.

- Figure3b would be more clear if the authors stacked the distributions rather than overlap

Figure 3b and corresponding supplementary figure 11 have been modified accordingly.

Reviewer #3 (Remarks to the Author):

Sallé et al. The global diversity of a major parasitic nematode is shaped by human intervention and climate

This carefully written paper focusses on the genomic variability of *Haemonchus contortus* from across the globe. The analyses are extensive and detailed, and the results have implications for understanding anthelmintic resistance. The paper is easy to follow and the English is excellent.

This reviewer can find no major flaws in the interpretation of results. The paper will be of general interest, and of more specific interest to medical, veterinary, and parasitology fields.

We thank the reviewer for their appraisal of our work.

In general, the first half of the results, dealing with the genetic variability and possible migration patterns, is not as interesting or novel as the results on variability associated with drug resistance and adaptation to climate. That is, the first half contains results that mainly confirm what is expected from the sample size, distance between sampling sites, and known history of human movement, and the results mainly 'confirm' expectations. Therefore, this reviewer would recommend shortening the sections dealing with genetic variation and migration, and save the space for more details about the much more important results in the following sections.

Following the first reviewer's comments, we have better highlighted the genes of interest. However, the questions raised by the second referee made it impossible to shorten sections about migration. Additional simulation analyses have been performed that brought relevant insights and significant modifications to our previous findings and understanding of *H. contortus* history. To this regard and to fit the requested concision in the formatting, we stuck to the original weight given to these sections.

This reviewer has several concerns about the sampling design and definition of 'population':

Line 652: Considering the distribution of this parasite, and of the sampling area, 267 individuals is not a large sample. 'Taking advantage of an available collection' is not motivation for a sampling regime.

Please provide more details on the choice of these individuals and how they are deemed to cover most (or which part) of genomic variability.

The sampling regime was motivated to delineate the contribution of major evolutionary forces, *i.e.* migration and selection (drug, climate). Because migration was likely to match human history, isolates from western African countries and West Indies were selected to address the contribution of slave trade history to *H.contortus* populations, and former colonies of the British Empire (South-Africa, Australia) were sampled to establish the connectivity between worm populations from these countries. Ivermectin-resistant isolates were also retained to evaluate how anthelmintics had shaped *H.contortus* genomic variability. Isolates were selected based on available material to ensure minimal sample size (n = 9) per isolate for proper allele frequency estimation. Following these criteria, populations from 12 countries were available. We accept that the available samples do not represent an ideal sample collection, but we are constrained by the availability of parasite material from other regions.

To address this comment, we have expanded on the description of sampling in the materials and methods section.

Line 654: The authors claim that 19 populations were sampled, but what is the definition of population here? Perhaps 'sampling site' would be sufficient?

Population refers to field isolate, collected from one sampling site (farm) in each case.

To address this remark, this definition has been brought to the manuscript under the materials and methods section. The term "*population*" was replaced with "*isolate*" throughout the manuscript.

Line 656: Metadata is said to be detailed in Supplementary Table 1, but this table only provides country names for most 'populations', while the 'populations' in France are equal to farm. Do we assume that each of the other 'populations' in other countries is equivalent to one farm? More details should be provided about the sample origins in Suppl. Table 1, otherwise it is difficult to tell what kind of 'populations' are being compared. If samples from each country come from smaller (farms) or larger areas (we cannot tell for most countries), this could change the interpretation of the results.

Each isolate was sampled from one farm unless stated otherwise.

This precision has been brought to the supplementary table legend.

Second revision

Dear Editor,

We would like to apologize for the discrepancies found in our previous response to referees. These mistakes have now been corrected. Please find attached our point-by-point response to the referee's comments below (line numbering refers to the tracked version of the revised manuscript).

Dear Dr Sallé,

Your manuscript entitled "The global diversity of the major parasitic nematode *Haemonchus contortus* is shaped by human intervention and climate" has now been seen by 1 referees. You will see from their comments below that while they find your work of interest, some important points are raised. We are interested in the possibility of publishing your study in Nature Communications, but would like to

consider your response to these concerns in the form of a revised manuscript before we make a final decision on publication.

We therefore invite you to revise and resubmit your manuscript, taking into account the points raised. In case you are unable to address concerns regarding effect of climate change on genetic variation raised by reviewer #3 by further analyses, please make sure to acknowledge and discuss them in the manuscript and tone down conclusions in this regards. Please make sure that the revised manuscript addresses these and all other remaining concerns in full and highlight all changes in the manuscript text file.

We are committed to providing a fair and constructive peer-review process. Do not hesitate to contact us if you wish to discuss the revision in more detail or if there are specific requests from the reviewers that you believe are technically impossible or unlikely to yield a meaningful outcome.

Reviewers' comments:

Reviewer #2 (Remarks to the Author):

The author's did a commendable job addressing our comments. We still require some clarifications to the below comments.

Key:

R2: Reviewer 2

A: Author response

R2A: Reviewer 2 response

R2: how were you able to reach the conclusion in line 190? Lower effective population sizes on islands could lead to more drift or smaller founding populations

--

A: Insular populations usually suffer higher exposure to drift or derive from smaller founding populations. Both factors are expected to reduce within population genetic diversity which in turn inflates F_{ST} estimates (Charlesworth 1998, *Mol. Biol. Evol.* 15: 538–543). We hence assumed that higher F_{ST} observed for this population was reflecting their insular situation and likely exposure to higher drift. Inferences from joint demography (dadi) between STO and other populations suggest isolation occurred in the XVII th century, despite a more complex scenario with MOR population.

--

R2A: To clarify, our comment was aimed at the assumption that high F_{ST} was due to isolation. High F_{ST} , as noted in your response can be due to small founding populations or high drift, likely confounding. However, F_{ST} is a measure of drift and not really a good estimator of migration (Whitlock and McCauley 1999). So we interpreted isolation to mean that you were assuming no migration. Your response is adequate, however, we can not find the quoted statement in the text. We can only locate "Inference of the joint history of STO and other African population supported this view." There is no mention of "secondary contact" in manuscript.

Following latest rewording of the manuscript, it appears the original statement (the one provided in our response) was modified to (L197-203): *"Inference of the joint history of STO and other African populations supported this view. The best-fitting model (supplementary Table 3) was consistent with either ancient symmetrical gene flow followed by isolation (since 1560 at least) or an early split followed by secondary contact in late 1800's before isolation with MOR population (supplementary Table 3). This latter demography would underpin the pattern of admixture detected between STO and MOR (Fig. 2c)."*

We apologize for this discrepancy. To make this statement clearer we inserted "as a result of higher exposure to drift or smaller founding populations" (L198) and added "in late 1800's" (L203).

R2: Can the authors also comment on the influence of population mean coverage on these tests since it was found that nucleotide diversity was affected by coverage.

A: We are not sure about which tests the reviewer is referring to and assume this is Tajima's D. A short mention on the impact of coverage for Tajima's D estimation had been provided in supplementary Figure 17. We found that Tajima's D was slightly higher with higher coverage, but chromosome-wide patterns were similar.

R2A: Yes, we were referring to TajD as your results indicated an affect on estimates of the pairwise diversity (π). If the TajD value is more positive with higher coverage (possible more accurate estimates of π), will it also be biased to be more negative in lower coverage regions?

We agree that local depth of coverage could bias our analysis but our data indicate that this phenomenon is not occurring. We retrieved the local depth of coverage using vcftools for the 23 million SNPs identified and computed a windowed average coverage for the three populations with highest coverage (FRA, GUA, NAM). Correlation between average window coverage and the respective estimated Tajima's D was weak in all cases (Pearson's $r = -0.17, -0.13$ and -0.04 for FRA, GUA and NAM populations). This new result has been provided in supplementary Figure 20 and the following has been inserted under the materials and methods section (L 919): "*A slight increase in Tajima's D values was observed in the same samples following resequencing, i.e. increase in depth of coverage. However, only modest and negative correlations were found between window coverage and Tajima's D estimates suggesting that coverage is not causing a systematic bias in estimates of Tajima's D (supplementary Fig. 20).*"

R2A: The authors agreed that introgression is not the proper terminology and are said to have replaced it, yet it appears in the manuscript Lines: 621, 1021, 1027, 1035

These were indeed overlooked. To account for this, the following modifications have been brought:

- L318: "genetic material"
- L635: "*back-cross* experiment"
- L1066: "The *introduction* of resistant haplotypes from France mainland into Guadeloupian"
- L1071: "ultimately providing inference of *gene flow* events in discrete"
- L1080: "to ensure that evidence of *gene flow* was consistent in both cases."

Supplementary material was also updated.

R2A: The authors said to have changed FRG to GUA, but FRG is still present in much of the manuscript and having FRG and GUA refer to the same thing is quite confusing. Lines 296, 299, Fig 4A, Fig 1A

We apologize for this issue. This has now been fixed with the appropriate Figure 1A being inserted in main text and corrections have been brought to lines 296 and 299 (now 308 and 311). Supplementary figures 1, 3, 4, 6, 10 and 23 and their respective captions have also been updated.

R2: line 528, backgrounds independently could be due to shared ancestry and standing variation.

A2: To address this comment, we have rephrased this line to: "The co-occurrence of resistant haplotypes in disconnected isolates could result from selection acting upon variants appeared in several backgrounds independently 51 , or from shared ancestry as found between France and Guadeloupe resistant individuals. This latter observation"

R2A: The sentence does not appear in the manuscript as there is no mention of independent backgrounds. Instead Line 593 reads: "The co-occurrence of the same resistant haplotypes in geographically disconnected isolates is almost certainly due to the independent evolution of benzimidazole resistance as has been described previously". I dont understand how the authors can claim that it is certainly due to independent evolution without testing the hypothesis with their data.

To clarify this, we have built a neighbour-net graph based on allelic divergence observed for the 296 SNPs called over the beta-tubulin locus between the 74 samples with sufficient coverage. This network shows resistant populations are polyphyletic, consistent with independent origin of beta-

tubulin mutations. This was even true within the French population (FRA.1) that was split into two subclades. However, the clustering of FRA.1 and FRG reflected the already described shared ancestry. The heatmap of allelic differences between samples over the beta-tubulin locus was replaced with the split network to make our point clearer (supplementary Figure 10). A sentence was added to the result section (L295): “A phylogenetic network based on genotype information over the whole *Hco-tbb-iso-1* locus (Supplementary Fig. 10) revealed that resistant populations were polyphyletic suggesting resistant mutations have generally evolved independently in different populations, e.g. Namibia (NAM) and South-Africa (STA.1, STA.3) and to a lesser extent in Australia, however French and Guadeloupian individuals had little divergence in their haplotypes, suggesting that gene flow of resistant haplotypes between mainland France and the West Indies may have occurred (Supplementary Fig. 10).”.

Further comments:

The climate change influencing genetic variation is still problematic since:

1) Only 3 comparisons

We originally focused on these three populations as they had the best coverage. Nevertheless, comparisons between additional populations (with a mean coverage above 3 and at least 5 individuals sequenced) also yield the same differentiation signal on chromosome III (supplementary figure 15). In every case but between Namibia and Indonesia, the genomic windows overlapping the *Pc* ortholog locus were departing from the F_{ST} distribution. These additional results have been provided as a supplementary Figure 15 and commented under the result section (L451): “Additional comparisons between populations with different histories, i.e. Australia, Indonesia, São Tomé and Zaire also supported this signal (supplementary Figure 15) suggesting it was independent of the precise choice of population included.”

2) No account for shared population history when identifying the loci

The fact that this differentiation signal holds across a wide range of comparisons performed would favour its relative independence from population ancestry. In addition, differentiation signal was found between populations with common ancestry, e.g. Namibia and Zaire or Australia and Indonesia; this suggests that differential ancestry is not confounding this finding.

3) The method of using a PCA to select from highly correlated variables, does not remove the correlation.

We agree that our PCA approach has not removed all of the correlations between the climate variables we model, but do not think this has had a major effect on the conclusions of this work.

Firstly, the PCA approach, identifying a small number of climatic variables representing each highly co-linear group of variables, has removed much of the correlation present.

Secondly, we have used a random forest algorithm that attempts to account for correlation between variables by computing variable importance using a conditional permutation on correlated variables, that is intended to counteract the “correlation bias” inherent in these methods (Tolosi et al. 2011, Bioinformatics). While this “correlation bias” is expected to result in the contribution of groups of correlated variables being downweighed, in fact we see remaining correlations between BIO7, BIO4 and BIO12 – variables that are consistently ranked as the most important features – suggested that correlation bias is having little effect on the limited number of features considered in our analysis.

Finally, we have further investigated this by performing hierarchical average clustering on the variables that remained after our PCA based variable selection to explore remaining correlation structure (an approach followed by Huang et al. 2003, The Lancet (cited in Tolosi et al. 2011)). This approach identified 5 main clusters composed of BIO4, BIO12, BIO19, BIO2 and BIO15, and a last cluster containing BIO7, BIO5, BIO9 and BIO10. We ran a random forest analysis on the subset of variables from these clusters (BIO4, BIO12, BIO19, BIO2 and BIO7) and found BIO12 and BIO7 as

the most important features. The same conclusion was found with BIO15 instead of BIO2. Our analysis identifying the importance of BIO12 and BIO7—thus seems robust to the details of variable selection.

These details were added in supplementary Fig. 16c and under the methods section (L1136):” *Further variable selection following hierarchical average clustering yielded similar outcomes, suggesting remaining correlations between variables were not affecting the random forest algorithm (supplementary Fig. 16c).*”.

We have also modified the discussion to make it clear that these variables stand for a group of highly correlated climate variables and to recall that none of our approaches is ideal (L. 645): “*Our analyses identified associations between genetic variation and bioclimatic variables on the one hand, and estimated genome-wide genetic differentiation between populations experiencing contrasting climates. Neither of these two analyses are ideal, as bioclimatic variables are correlated and genetic differentiation is influenced by confounding factors such as population ancestry. Nevertheless, the correlation between bioclimatic variables was expected to lead to our analysis underestimating the respective contribution of the most important features, but BIO12 and BIO7 were consistently ranked as the most important features. Genetic differentiation was conserved across various comparisons suggesting the reported signal was relatively independent of population ancestry.*”

4) Reference *Drosophila* for all gene ontologies

Our gene ontology enrichment analyses were based on the gene ontology terms assigned to gene models in the *H. contortus* v1.0 genome. These terms were assigned using WormBase GO terms via orthology with *C. elegans* and from interproscan protein domains.

Our discussion of *Drosophila* is because there is evidence for the function of our *Pc* ortholog candidate gene in that species. While in the absence of direct experimental evidence we cannot be definitive, we believe it is at least plausible that this function is conserved:

- domain prediction from its peptide sequence indicates a chromo- or chromo shadow domain, supporting a role in chromatin-binding.
- One-to-one orthology between *HCOI01540500* and *Pc* in *Drosophila melanogaster* was originally found in WormBaseParasite. Current data from the WormBaseParasite database (current identifier *HCON_00087060*) suggest that no one-to-one ortholog of this gene was found in *C. elegans* but present in 11 other clade V nematode species including two free-living organisms (*O. tipulae*, *C. angaria*) and 9 parasitic species (*A. ceylanicum*, *A. duodenale*, *D. viviparous*, *H. placei*, *H. polygyrus*, *N. americanus*, *N. brasiliensis*, *O. dendatum*).
- In addition, derived orthologs in mammals (chromobox, *cbx*) and a putative homolog in *C. elegans* (*cec-1*) was reported by Whitcomb et al. Trends in Genetics 2007. More recent functional insights (Saltzman et al. Genetics 2018) seem to support this homology with evidence that the repressive Polycomb-specific H3K27me methylation mark was recognized by the CEC-1 protein (and CEC-6, another chromo domain protein).

Some of these elements have been inserted under the discussion section to emphasise that functional evidence is still missing but supported by other insights (L. 670-677): “*While in the absence of direct experimental evidence we cannot be definitive, we believe it is possible that this function of this *Pc* ortholog is conserved in nematodes. Domain prediction from its peptide sequence indicates a chromo- or chromo-shadow domain supporting a role in chromatin-binding. In addition, the putative *C. elegans* homolog of this gene⁵⁹, named *cec-1*, encodes a protein that recognizes the repressive Polycomb-specific H3K27me methylation mark⁶⁰. Knocking out the expression of this gene through RNA interference should help to support its role in the plastic response to climatic stress.*”

Final referees' comment

Reviewer #2 (Remarks to the Author):

No further comments. Thank you for addressing the additional comments and concerns.